# A small number of workers with specific personality traits perform tool use in ants

István Maák[1,2], Garyk Roelandt[3], Patrizia d'Ettorre[3,4]*

[1]Department of Ecology, University of Szeged, Szeged, Hungary; [2]Museum and Institute of Zoology, Polish Academy of Science, Warsaw, Poland; [3]Laboratory of Experimental and Comparative Ethology UR 4443, University Sorbonne Paris Nord, Villetaneuse, France; [4]Institut Universitaire de France (IUF), Paris, France

**Abstract** Ants use debris as tools to collect and transport liquid food to the nest. Previous studies showed that this behaviour is flexible whereby ants learn to use artificial material that is novel to them and select tools with optimal soaking properties. However, the process of tool use has not been studied at the individual level. We investigated whether workers specialise in tool use and whether there is a link between individual personality traits and tool use in the ant *Aphaenogaster senilis*. Only a small number of workers performed tool use and they did it repeatedly, although they also collected solid food. Personality predicted the probability to perform tool use: ants that showed higher exploratory activity and were more attracted to a prey in the personality tests became the new tool users when previous tool users were removed from the group. This suggests that, instead of extreme task specialisation, variation in personality traits within the colony may improve division of labour.

*For correspondence:
d-ettorre@univ-paris13.fr

**Competing interests:** The authors declare that no competing interests exist.

## Introduction

Tool use is a widespread phenomenon within the animal kingdom (*Shumaker et al., 2011*; *Sanz et al., 2013*) and new examples of animal tool use are regularly discovered, such as recently in pigs (*Root-Bernstein et al., 2019*) and seabirds (*Fayet et al., 2020*). Tool use is defined as "the external employment of an unattached or manipulable attached environmental object to alter more efficiently the form, position, or condition of another object, another organism, or the user itself, when the user holds and directly manipulates the tool during or prior to use and is responsible for the proper and effective orientation of the tool" (*Shumaker et al., 2011*, p. 5). Most of the reports concern vertebrates, particularly primates and birds, which can manufacture tools to solve specific tasks (*Hunt and Gray, 2002*; *Sanz et al., 2013*; *Auersperg et al., 2014*), use multiple tools sequentially (*Martin-Ordas et al., 2012*), and choose effective tools based on their functional properties (*Visalberghi et al., 2009*). We know that capuchin monkeys have been using stone tools to process food for at least 3000 years (*Falótico et al., 2019*) but presumably the use of tools appeared even earlier in invertebrates. *Smith and Bentley-Condit, 2010* reported about 50 cases of tool use in insects, encompassing 30 different genera. Among the best described examples is the use of debris to transport liquid food by some species of ants (*Morrill, 1972*; *Barber et al., 1989*), in particular, several species of the genus *Aphaenogaster* (*Fellers and Fellers, 1976*; *Tanaka and Ono, 1978*; *McDonald, 1984*; *Agbogba, 1985*). Workers of these species are characterised by the lack of a distensible crop and by a chitinous gaster, preventing the transportation of large amounts of liquid food inside their bodies, a feature common in many other ant species (*Hölldobler and Wilson, 1990*; *Davidson et al., 2004*). Moreover, unlike most ant species, *Aphaenogaster* workers do not perform mouth-to-mouth exchange of liquid food (i.e. trophallaxis, *Delage and Jaisson, 1969*); therefore, foragers cannot use this form of food transmission among colony members. Workers forage individually mostly on the ground level and can cover large areas in habitats with scarce food

sources (*Cerdá et al., 1998*). When *Aphaenogaster* foragers discover a source of liquid food, such as fruit pulp or body fluids of dead insects, they collect debris (pieces of leaves, soil, sand grains), drop them into the liquid food, and then transport these soaked debris to their nest. This behaviour qualifies as tool use, namely tool-assisted food transport. Indeed, these ants do not drop debris in non-food substances (*Banschbach et al., 2006*).

In a previous study (*Maák et al., 2017*), we have investigated whether *Aphaenogaster* ants are selective in the choice of material to be used as tools and we demonstrated that ant workers prefer materials that are easy to handle and with good soaking capacity. Furthermore, ants can learn to use artificial material that is novel to them and select the material with optimal soaking properties, thus showing that tool use is not behaviourally fixed in ants (*Maák et al., 2017*). Tool selection also depends on the foraging environment and varies with food type (viscosity), distance, and availability of tools (*Lőrinczi et al., 2018*).

The process of tool use in ants has not been studied at the individual level. In particular, we do not know whether any forager with sufficient information about the location of the food and the availability of the tools would perform tool use or whether there are specialised workers which perform tool use repeatedly. In *A. rudis*, it was observed that the tool use behaviour is carried out by a small subset of individuals within the group of foragers and only a small number of workers perform the debris dropping task (*Banschbach et al., 2006*). In *A. subterranea* as well, the proportion of workers observed to use tools was only a small fraction of the total number of foragers. The number of tool users did not increase with colony size, while the number of total foragers did. Moreover, there was no significant relationship between the number of ants working at the debris dropping task and the number of debris pieces dropped, indicating that a small number of workers can perform this task very efficiently (*Lőrinczi, 2014*).

We asked the question of what makes a good tool user in *A. senilis* ants. It is indeed unknown whether some individual characteristics of workers, such as personality traits, would predict tool use behaviour. When inter-individual differences in behavioural traits are consistent across time and/or context they are considered personality traits (*Réale and Dingemanse, 2012*). In ants, both workers and colonies can show personalities (*Pinter-Wollman, 2012*) and there is a link between individual and collective behaviour (*Carere et al., 2018*). There is some evidence that the allocation of workers to a certain task may be influenced by individual personality, for instance in the ant *Leptothorax acervorum* (*Kühbandner et al., 2014*), where more exploratory and aggressive workers were also more active in the nest (rested less). This inter-individual difference could enhance division of labour (*Jeanson and Weidenmüller, 2014*), the phenomenon in which different individuals perform different tasks. Division of labour is beneficial for colony organisation and it is believed to be one of the principal factors explaining the extraordinary ecological success of social insects (*Hölldobler and Wilson, 1990*). A broadly acknowledged model explaining the emergence of division of labour within a colony is the 'response threshold model', which posits that individual workers differ in their sensory perception and/or in their behavioural responses to stimuli associated with specific tasks (*Robinson, 1992*; *Beshers and Fewell, 2001*). However, the possible interplay between consistent individual differences and division of labour has not been explored (*Jeanson and Weidenmüller, 2014*).

We also know that there is an association between personality traits and cognitive traits in ants. Consistent individual differences in exploratory activity predicted learning performance of individual carpenter ants, with 'active-explorers' being slower in learning than 'inactive-explorers' (*Udino et al., 2017*) and individual differences in exploratory activity were linked to cognitive judgement bias, the propensity to anticipate either positive or negative consequences in response to ambiguous information (*d'Ettorre et al., 2017*). Therefore, we expected a link between individual personality traits and tool use.

We performed three different experiments to study the tool use behaviour at the individual level in the ant *A. senilis*. In Experiment 1, we observed the tool use process in whole colonies to characterise the behaviour of ants during the two parts of the process: the transport of tools to the bait and the transport of tools from the bait to the nest. In social insects, task partitioning -a phenomenon in which a piece of work is divided among two or more workers- has been reported in a wide variety of foraging tasks in several species of ants, bees, wasps and termites (*Ratnieks and Anderson, 1999*). However, partitioning may reduce task efficiency and reliability unless the number of workers involved in the task is high or a given subtask is carried out by morphologically specialised

workers (*Ratnieks and Anderson, 1999*). In *Aphaenogaster*, workers are morphologically similar, therefore we predicted that we would not observe task partitioning and that the same worker could perform both parts of the task.

In Experiment 2, we created sub-colonies to investigate whether there is an individual specialisation in tool use. Are the same ants using tools repeatedly? Are ants that use tools to collect liquid food also transporting solid food items? We predicted that ants would specialise in tool use only, under the general assumption that specialised individuals work more efficiently than less specialised ones (*Robinson, 1992*). In this experiment, we also asked whether the simple fact of observing nest-mates using tools during one trial would facilitate the task performance of naive workers by social learning. The involvement of social learning in solving complex tasks has been shown in social insects, for instance, bumblebees (*Loukola et al., 2017*).

In Experiment 3, we studied the possible link between individual personality traits and tool use. Previous studies demonstrated the existence of colony-level personality in *A. senilis* (*Blight et al., 2016a*; *Blight et al., 2016b*), we thus predicted that consistent behavioural differences will be found also at the individual level in this species (worker personality). Using workers that were individually characterised for personality traits, namely exploratory activity and reaction to prey, we created sub-colonies that were tested in tool use trials. After each trial, the ants that used tools were removed from the sub-colony. Thus, we could test whether personality would predict which ants will become tool users in the next trial.

## Results

### Experiment 1: tool use process in whole colonies

To characterise the entire tool using process, the plastic box housing the ant colony was connected to a foraging arena with a detachable bridge. At the beginning of the experiment, we placed 10 tools (small pieces of sponge) and then liquid food (diluted honey) in the foraging arena. Next, we removed the bridge so that ants in the foraging arena could not go back in the ant colony. This way we could quantify the number of workers transporting tools to the bait over the total number of workers that were present in the foraging arena. After 30 min, the connecting bridge was replaced, giving the possibility to the ants to transport tools from the bait to the nest (see Materials and methods). We used three colonies and five replicates each (one replicate = 1 trial with food and tools). The results show that the number of workers present in the foraging arena before replacing the bridge had a positive effect on the latency to drop the first tools into the bait (LMM $t = -2.22$, $N = 15$, p=0.047), meaning that the higher the number of workers present, the shorter the latency. Also, the shorter was the latency to drop the first tool into the bait, the shorter the total time needed to transport all the tools to the bait ($t = 2.26$, $N = 15$, p=0.043). However, the number of workers involved in tool use did not have any effect on the latency to drop the first tool to the bait ($t = 1.34$, $N = 15$, p=0.20) nor on the total time of tool transport to the bait ($t = -0.46$, $N = 15$, p=0.65).

The latency to transport the first tool inside the nest did not depend on the dynamics of the tool transport to the bait. In particular, the latency for the first tool transport to the nest was not influenced by the latency to drop the first tool into the bait (LMM $t = 0.99$, $N = 15$, p=0.34), nor by the total time to transport all tools to the bait ($t = -1.35$, $N = 15$, p=0.21), nor by the number of workers involved in tool transport to the bait ($t = -0.44$, $N = 15$, p=0.67). The transport to the nest started always well after (124.4 ± 15.3 min, mean ± SE; *Table 1*) the completion of the tool transport to the bait.

We then investigated further the workers that used tools. Compared to the number of workers present in the foraging arena (23.33 ± 6.04 workers, mean ± SE), only a few workers performed the tool use behaviour (2.33 ± 0.35 workers, mean ± SE; *Table 1*). We observed workers repeatedly transporting tools within the same trial and between trials (*Table 2*). Some workers participated in both parts of the task (transport of tools to the bait and also from the bait to the nests) and repeated transports by the same worker were observed in both parts of the tool use process (*Table 2*). In the next step, we asked whether the tool users were simply those workers that were the first to discover the location of the food and the tools. This was not the case. Indeed, an average of 11.35 ± 6.49 workers (mean ± SE, min = 3, max = 33 workers; see *Table 3* for further details)

**Table 1.** Experiment 1: tool use process in whole colonies.

Summary table showing the number of workers present in the arena (# workers in arena), the latency to drop the first tool on the bait (First tool on bait), the total time devoted to tool transport to the bait (Tot. time tool transport), the number of tools transported to the bait (# tools on bait), the latency to transport the first tool to the nest from the start of the experiment (First tool to nest) and the number of workers involved in tool transport to the bait (# workers transp. tools to bait). Five replicates (R1-R5) for each colony are shown. The last column shows the number of tools transported by each worker to the bait (# tools transp. by each worker); for instance, in R1 there were two tool users, one transported nine tools and the other 1.

| Colony | # Workers in arena | First tool on bait (min) | Total time tool transport (min) | # Tools on bait | First tool to nest (min) | # Workers transp. tools to bait | # Tools transp. by each worker |
|---|---|---|---|---|---|---|---|
| 1 R1 | 19 | 9 | 21 | 10 | 174 | 2 | 9; 1 |
| 1 R2 | 43 | 19 | 19 | 10 | - | 2 | 5; 5 |
| 1 R3 | 16 | 23 | 15 | 10 | 201 | 3 | 1; 8; 1 |
| 1 R4 | 23 | 13 | 37 | 9 | - | 2 | 7; 2 |
| 1 R5 | 29 | 4 | 23 | 10 | - | 2 | 5; 5 |
| 2 R1 | 6 | 26 | 22 | 10 | 111 | 5 | 1; 2; 1; 4; 2 |
| 2 R2 | 14 | 7 | 20 | 10 | 96 | 2 | 2; 8 |
| 2 R3 | 40 | 1 | 15 | 8 | - | 3 | 3; 1; 4 |
| 2 R4 | 30 | 3 | 25 | 10 | 144 | 2 | 5; 5 |
| 2 R5 | 71 | 2 | 10 | 10 | 126 | 3 | 7; 2; 1 |
| 3 R1 | 16 | 45 | 20 | 10 | - | 3 | 1; 4; 5 |
| 3 R2 | 9 | 7 | 6 | 10 | 172 | 2 | 8; 2 |
| 3 R3 | 23 | 7 | 7 | 10 | 222 | 1 | 10 |
| 3 R4 | 4 | 18 | 24 | 10 | 178 | 1 | 10 |
| 3 R5 | 7 | 120 | 31 | 9 | 214 | 2 | 5; 4 |

contacted the tools and the food before the first tool user dropped the first tool into the bait. Paint marked tool users ($N$ = 18) took on average 388 ± 134.16 sec (mean ± SE) to locate the tools after contacting the food for the first time. Once they had the information about the location of both the food and tools, the latency to drop the first tool into the bait was 385.67 ± 128.30 sec (mean ± SE). However, these two latencies were not correlated ($r_s$ = −0.006, $N$ = 18, p=0.98), meaning that those workers that located the tools earlier did not necessarily start the transport of tools to the bait faster.

**Table 2.** Experiment 1: tool use process in whole colonies.

The tool use behaviour is composed of two parts: transport of tools to the food source (bait) and transport of imbibed tools inside the nest. The table shows the total number of tool users that participated (Particip.) in both parts of the tool use process and those that transported tools to the bait (# tool users), tool users that were marked (Marked tool users), number of workers that transported more than one tool within a trial (>1 tool within trial), and that transported more than one tool among trials (>1 tool among trials). Shown is the total for the three experimental colonies (sum of five trials each).

| | | Transport to bait | | | | Transport to nest | |
|---|---|---|---|---|---|---|---|
| Colony | # particip. in both parts | # Tool users | Marked tool users | >1 tool within same trial | >1 tool across trials | Marked tool users | >1 tool within same trial |
| 1 | 1 | 11 | 5 | 5 | 1 | 3 | 1 |
| 2 | 1 | 15 | 3 | 3 | 1 | 5 | 1 |
| 3 | 2 | 9 | 8 | 8 | 0 | 9 | 1 |

**Table 3.** Experiment 1: tool use process in whole colonies.

The tool users were not the first workers obtaining information about the presence of food and tools. The table shows the number of workers that contacted both the tools and the food before the first tool was dropped into the bait; the latency (Lat.) for the first worker to obtain information (info.) about the presence of both the tools and the food and the latency for the first tool user to obtain information about the tools and the food.

| Colony | # Workers contacting tools and food before the first tool was dropped on the bait | Lat. first worker having info. about both tools and food (min) | Lat. first tool user having info. about both tools and food (min) |
|---|---|---|---|
| 1 R1 | 10 | 2.61 | 7 |
| 1 R2 | 17 | 0.17 | 17 |
| 1 R3 | 11 | 0.55 | 23 |
| 1 R4 | 18 | 0.38 | 2 |
| 1 R5 | 8 | 0.7 | 3 |
| 2 R1 | 5 | 18.03 | 25 |
| 2 R2 | 5 | 0.98 | 7 |
| 2 R3 | 3 | 0.75 | 1 |
| 2 R4 | 4 | 0.87 | 2 |
| 2 R5 | 5 | 0.18 | 1 |
| 3 R1 | 27 | 0.75 | 45 |
| 3 R2 | 4 | 1.92 | 7 |
| 3 R3 | 11 | 1.05 | 7 |
| 3 R4 | 4 | 1.85 | 5 |
| 3 R5 | 33 | 1.75 | 106 |

## Experiment 2: is there specialisation in tool use?

In this experiment, we investigated whether ants specialise in tool use or also transport solid food items (here cricket legs). We also tested the effect of a familiarisation-trial (pre-trial) with food and tools on naive workers. In this experiment, we focused only on the transport of tools to the bait. For the pre-trial, a group of sub-colonies (sub-colonies 1) was familiarised with the food (diluted honey) and the tools (10 pieces of sponge) separately, while the other group (sub-colonies 2) was familiarised with the tools and the food simultaneously. Therefore, workers of sub-colonies two could perform the tool use behaviour while workers of sub-colonies one could not. Immediately after this pre-trial, workers that performed tool use in sub-colonies two were removed so that we could see whether the fact of observing nestmates using tools during this pre-trial would facilitate tool use in naive workers by social learning.

The day after, each sub-colony was tested with liquid food and 10 tools, then with five cricket legs, then again with liquid food and 10 tools. The procedure was repeated the following day and, after 2 days of rest, each sub-colony received liquid food and 20 tools (*Figure 1A*). This procedure to test for specialisation in tool use was the same for sub-colonies 1 and sub-colonies 2.

Of the 40 individually marked workers per sub-colony involved in this experiment, about 20% performed tool use during at least one of the four trials with 10 tools (*Table 4*). The average number of individuals per sub-colony showing tool use was 8.25 ($CI_{95\%}$ [6.47, 10.03]). This was notably lower than 10.2 ($CI_{95\%}$ [10.18, 10.22]), which is the average number of tool users obtained by randomly assigning the same number of observed tool use events to a simulated ant population based on the same number of individuals, sub-colonies and trials (see Simulation data of experiment 2, *Supplementary file 1*), and the confidence intervals do not overlap. This indicates that the individual distribution of tool use events was not random in our experiments. Of the 99 tool users in total, 64.7% performed several tool transports within the same trial and 33.3% participated repetitively in more than one trial (*Table 4*). The majority of these ants that performed multiple tool transport across trials (26 over 31 workers) participated in two trials (2.89 ± 0.42 per trial, mean ± SE), three workers participated in three trials and two workers in four trials (*Table 5*). It is important to note that the occurrences of tool use were repeatable across trials at the individual level ($R_{ICC} = 0.22$,

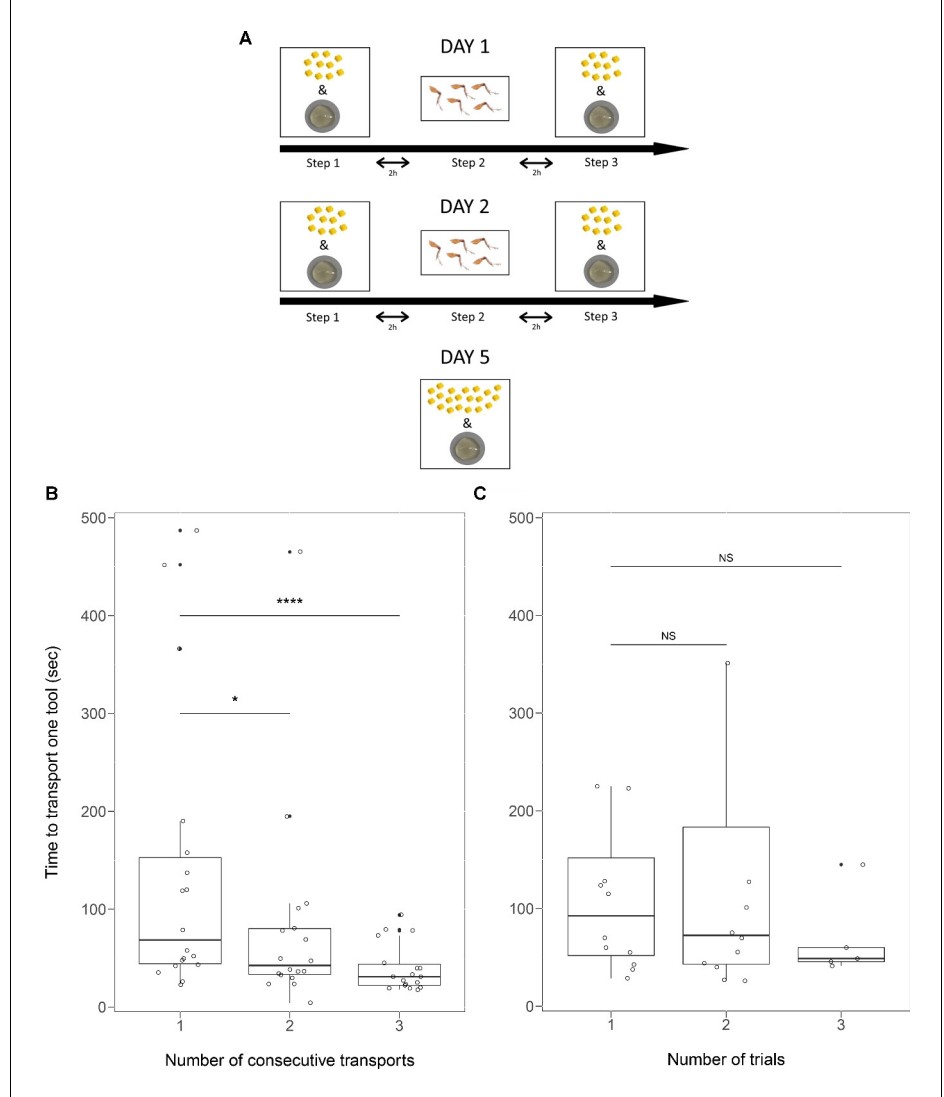

**Figure 1.** Procedure followed in experiment 2. Each sub-colony received honey and tools in steps 1 and 3. The yellow items represent the tools (small pieces of sponge,~1 mm3), below is shown the plate with diluted honey (0.25 ml). In step 2, the ants received five cricket legs (*Acheta domestica*). On day 5, the ants received 20 tools and the honey bait (**A**). For individual workers that transported four consecutive tools to the bait, the time needed to transport one tool significantly decreased within one trial (**B**). For individual workers that transported at least two consecutive tools in different trials, the time needed to transport one tool to the bait did not change among trials (**C**). Box plots show medians, quartiles, min-max values, outliers (black dots) and individual data points (empty circles). *NS* – non-significant, \*: p<0.05, \*\*\*\*: p<0.0001.

The online version of this article includes the following source data for figure 1:

**Source data 1.** Experiment 2 - the time (sec) needed to transport one tool by the same individual that performed consecutive tool transports to the bait within a single trial.

**Source data 2.** Experiment 2 - the time (sec) needed to transport one tool for individual workers that transported at least two consecutive tools in different trials.

*N* = 1880, p=0.001). Half of the workers (50.38%) that performed tool use in the final trial with 20 tools, also participated in at least one of the previous trials with 10 tools (*Table 6*), thus confirming that the same ants use tools over and over again, including across trials.

Workers that performed tool use were not necessarily specialised in foraging for liquid food; they also transported cricket legs to the nest. About half (48.31%) of the workers that transported cricket

**Table 4.** Experiment 2: Is there specialisation in tool use?
The total number of tool users and cricket leg transporters (transp.) and the percentage of workers that performed repeated tool use within or between trials or that participated also in the transport of cricket legs. The last column shows the number of very active workers, which participated in at least two tool use trials and one leg transport.

| Colony/ subcolony | Total tool users | Repeats within (%) | Repeats between (%) | Transporting also legs (%) | Total leg transp. | % tool users among leg transp. | ≥2 trials, ≥1 leg (%) |
|---|---|---|---|---|---|---|---|
| 1/1 | 6 | 4 (66.7) | 1 (16.7) | 2 (33.3) | 6 | 33.3 | 1 (16.7) |
| 1/2 | 4 | 3 (75) | 1 (25) | 1 (25) | 5 | 20 | 1 (25) |
| 2/1 | 12 | 7 (58.3) | 2 (16.7) | 1 (8.3) | 1 | 100 | - |
| 2/2 | 10 | 8 (80) | 4 (40) | 2 (20) | 5 | 40 | 2 (20) |
| 3/1 | 6 | 6 (100) | 5 (83.3) | 1 (16.7) | 7 | 14.3 | 1 (16.7) |
| 3/2 | 12 | 8 (66.7) | 3 (25) | 3 (25) | 3 | 100 | 2 (16.7) |
| 4/1 | 8 | 5 (62.5) | 2 (25) | 3 (37.5) | 6 | 50 | 1 (12.5) |
| 4/2 | 10 | 7 (70) | 4 (40) | 3 (30) | 6 | 50 | 2 (20) |
| 5/1 | 6 | 3 (50) | 3 (50) | 2 (33.3) | 4 | 50 | 2 (33.33) |
| 5/2 | 5 | 2 (40) | 2 (40) | 1 (20) | 4 | 25 | - |
| 6/1 | 14 | 8 (57.1) | 3 (21.4) | 4 (28.6) | 7 | 57.1 | 2 (14.29) |
| 6/2 | 6 | 3 (50) | 1 (16.7) | 2 (33.3) | 5 | 40 | 1 (16.7) |
| Average | 8.25 | 5.33 (64.7) | 2.58 (33.3) | 2.08 (25.9) | 4.92 | 48.31 | 1.25 (15.15) |

legs were also tool users in at least one trial (*Table 4*). Among the 99 tool users, 15 workers (15.15%) showed a particularly high activity by participating in at least two tool use trials and at least one cricket leg transport (*Table 4*).

The consecutive transport of several tools within one trial enhanced the efficiency of a tool using worker. When workers consecutively transported four tools, the time to transport one tool decreased with the number of tools transported (second-third tool: LMM $t = -2.25$, $N = 54$, p=0.03, second-fourth tool: $t = -3.75$, p<0.001; *Figure 1B*, *Source code 1*). However, the average time to transport one tool did not change significantly between different trials (first-second: $t = -0.01$,

**Table 5.** Experiment 2: is there specialisation in tool use?
Number of workers participating in more than one trial and the average number of tools they transported.

| Colony/ subcolony | Two trials | # Tools | Three trials | # Tools | Four trials | # Tools |
|---|---|---|---|---|---|---|
| 1/1 | 1 | 9.5 | | | | |
| 1/2 | | | | | 1 | 3.75 |
| 2/1 | 2 | 4 | | | | |
| 2/2 | 4 | 3.12 | | | | |
| 3/1 | 5 | 3.3 | | | | |
| 3/2 | 2 | 1.75 | | | 1 | 3.75 |
| 4/1 | | | 2 | 3 | | |
| 4/2 | 4 | 1.75 | | | | |
| 5/1 | 3 | 3 | | | | |
| 5/2 | 2 | 3.25 | | | | |
| 6/1 | 3 | 1.5 | | | | |
| 6/2 | | | 1 | 8.33 | | |
| Average | 2.89 | 3.46 | 1.50 | 5.67 | 1.00 | 3.75 |

**Table 6.** Experiment 2: is there specialisation in tool use?
The total number of workers using tools in the last trial (trial with 20 tools) and the number of workers that performed tool use also in previous trials (10 tools).

| Colony/subcolony | Total # tool users | # Using tools in previous trials (%) |
|---|---|---|
| 1/1 | 4 | 2 (50%) |
| 1/2 | 4 | 1 (25%) |
| 2/1 | 5 | 3 (60%) |
| 2/2 | 5 | 1 (20%) |
| 3/1 | 4 | 3 (75%) |
| 3/2 | 3 | 2 (66.7%) |
| 4/1 | 2 | 1 (50%) |
| 4/2 | 2 | 1 (50%) |
| 5/1 | - | - |
| 5/2 | 5 | 3 (60%) |
| 6/1 | 5 | 3 (60%) |
| 6/2 | 8 | 3 (37.5%) |
| Average | 4.27 | 2.09 (50.38%) |

$N = 29$, p=0.99; first-third $t = -1.02$, p=0.32; *Figure 1C*, *Source code 2*). It was possible that while a worker was involved in repeated tool transports to the bait, another worker started using tools. The interference of this new tool user did not alter the efficiency of the previous tool user (LMM $t = 0.5$, $N = 66$, p=0.62), meaning that the time to transport a tool to the bait by the first worker did not change before and after interference, and there was no significant effect of the sub-colony ($t = -0.52$, $N = 66$, p=0.6).

We analysed the effect of the single pre-trial -in which the sub-colonies two received liquid food and tools simultaneously, while the sub-colonies one received first food and then tools (in absence of the food)- on the performance of the ants in the subsequent trial (with food and tools at the same time). The latency to bring the first tool to the bait was not different between the two types of sub-colonies (LMM $t = -0.5$, $N = 12$, p=0.64; *Figure 2A*, *Source code 1*) as well as the total time of tool transport ($t = -0.88$, $N = 12$, p=0.42; *Figure 2B*, *Source code 2*). The number of workers involved in tool use was also not different between the two groups (LMM $t = -0.46$, $N = 12$, p=0.67), indicating that there is no social learning in these conditions.

## Experiment 3: does worker personality predict tool use?

We created eight sub-colonies each with 20 individually marked workers, which were characterised for personality traits using two tests, open-field and reaction to prey (*Figure 3A,B*), repeated after two days (see Materials and methods). Ants showed significant consistency over time in their behavioural responses. In the open-field test, the time spent walking in the periphery ($R_{ICC} = 0.27$, $N = 154$, $CI_{95\%}$ [0.11, 0.41], p<0.001) and the total time spent in the central area were significantly repeatable across the two sessions ($R_{ICC} = 0.43$, $N = 154$, $CI_{95\%}$ [0.31, 0.56], p<0.0001; *Figure 3C*, *Source code 1*). In the reaction to prey test, the time spent in contact with the prey was also highly repeatable ($R_{ICC} = 0.61$, $N = 154$, $CI_{95\%}$ [0.49, 0.71], p<0.0001; *Figure 3D*, *Source code 2*). The time spent in contact with the prey was negatively correlated with the time spent walking in the periphery ($r_s = -0.30$, $N = 154$, p<0.001) and positively correlated with the total time spent in the central area ($r_s = 0.33$, $N = 154$, p<0.001), indicating that the more exploratory ants were also those more interested in the prey.

Ants were observed in seven tool use trials (each with liquid food and 10 tools) over 4 consecutive days (*Figure 4*). After each trial, the ants that used tools were removed from the sub-colony. In this way, we could test whether personality would predict which ants will become tool users in the next

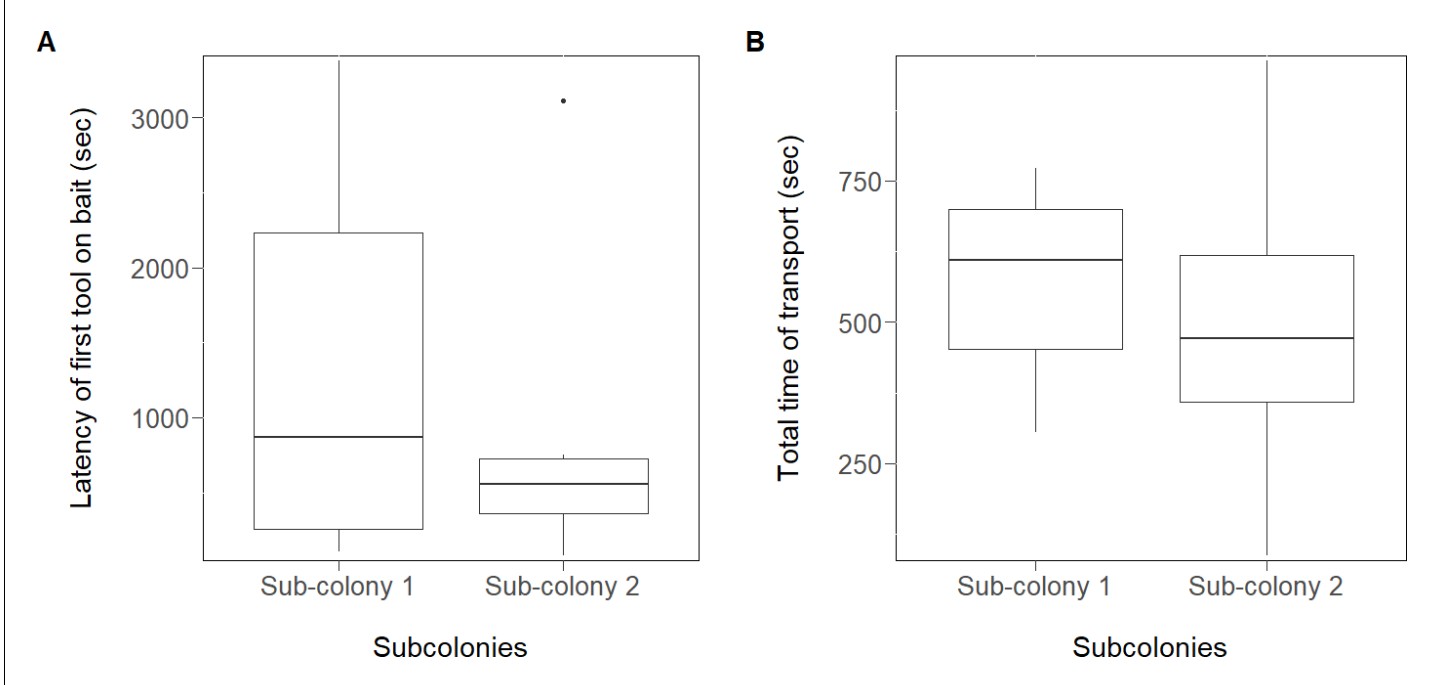

**Figure 2.** Effect of the pre-trial in experiment 2. There was no significant difference between the different groups of sub-colonies in the latency to transport the first tool to the bait (**A**) and the total time needed to transport the 10 tools to the bait (**B**). Sub-colonies 2: had a pre-trial (liquid food and tools simultaneously); Sub-colonies 1: without a pre-trial (received first food and then tools in absence of the food). Workers manipulating tools during the pre-trial were removed (see Materials and methods). Box plots show median, quartiles, min-max values and outliers (black dot).

The online version of this article includes the following source data for figure 2:

**Source data 1.** Experiment 2 - the latency (sec) to transport the first tool to the bait by the two subcolonies with and without a pre-trial.

**Source data 2.** Experiment 2 - the total time (sec) needed to transport the ten tools to the bait by the two subcolonies with and without a pre-trial.

trial. We used Principal Component Analysis based on the variables of the personality tests to calculate a 'personality score' (see Materials and methods and *Figure 5A,B*, *Source code 1*).

The ants' personality score significantly predicted the probability of using tools in at least one trial (GLMM $z = 3.97$, $N = 154$, p<0.001). Moreover, the confidence intervals of the personality score of tool users (mean = 0.57, $CI_{95\%}$ [0.26, 0.88]) and non-tool users (mean = −0.33, $CI_{95\%}$ [−0.56,–0.09]) did not overlap (*Figure 5C*, *Source code 2*). A positive personality score characterised those ants that spent more time exploring the central area in the open field test and that spent more time in contact with the prey (*Figure 5A,B*).

Given that tool users were more exploratory than non-tool users, one may wonder whether the observed relationship between personality and tool use behaviour is merely the result of the fact that more exploratory ants are more likely to encounter the tools and possibly use them up before other workers can find them. We thus analysed a randomly chosen subset of the tool use trials of experiment three to investigate this. First, it should be noticed that the tool use process did not start immediately after the tools and the food were offered to the ants. The latency to bring the first tool to the bait was 894.71 ± 153.49 sec (mean ± SE, $N = 21$ trials; see 'Contact_with_tools' in data of experiment 3, *Supplementary file 1*), which gives plenty of time for all the ants to explore and find the tools. From a total of 21 trials, only in five occasions, the tool user was the first worker to investigate the tools. In the majority of the trials (16 out of 21) an average of about five workers investigated the tools by antennation before the first tool user but did not pick them up (4.94 ± 0.52 (mean ± SE) workers, min = 1 worker, max = 8 workers). This indicates that the tool use behaviour is not merely prompted by the fact of finding the tools or the food before other workers. If this would be the case, in the reaction to prey test we should expect a negative correlation between the latency to find prey and the time spent in contact with it (the shorter latency, the longer contact time). In

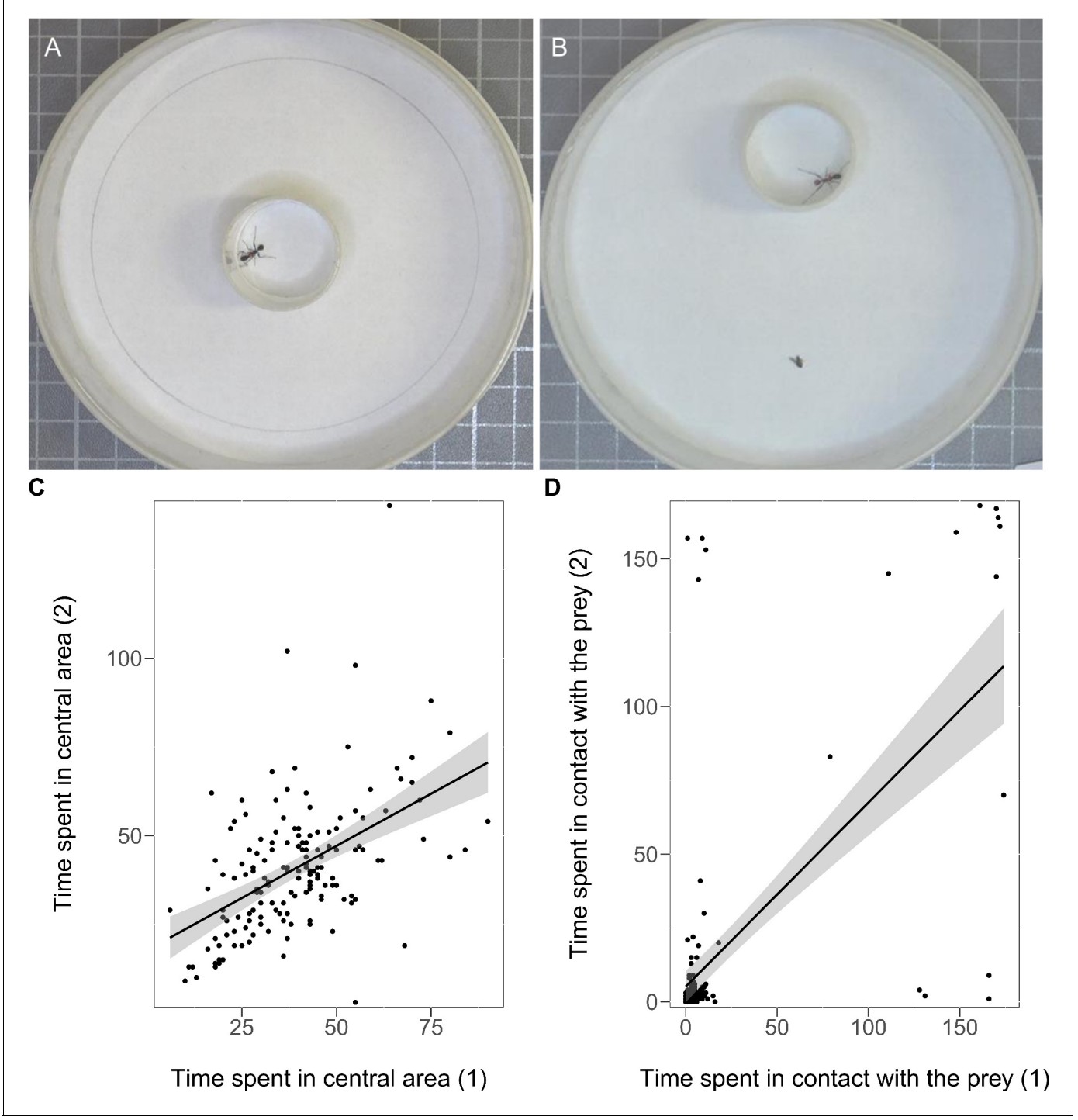

**Figure 3.** Experimental set-up used for the open-field (**A**) and the reaction to prey tests (**B**). The ant is in the acclimatisation tube for 1 min and the test starts when the tube is removed (see Materials and methods). The consistency across the two sessions of the open-field test regarding the time spent in the central area (**C**) and the consistency across the two sessions of the reaction to prey test (**D**). The black line with confidence band (grey) is plotted based on the Pearson correlation of the two variables.

The online version of this article includes the following source data for figure 3:

**Source data 1.** Experiment 3 - correlation between the two sessions (repeats) for the time (sec) spent walking in the central area (open-field test) by individual ants.

**Source data 2.** Experiment 3 - correlation between the two sessions (repeats) for the time (sec) spent in contact with the prey (reaction to prey test).

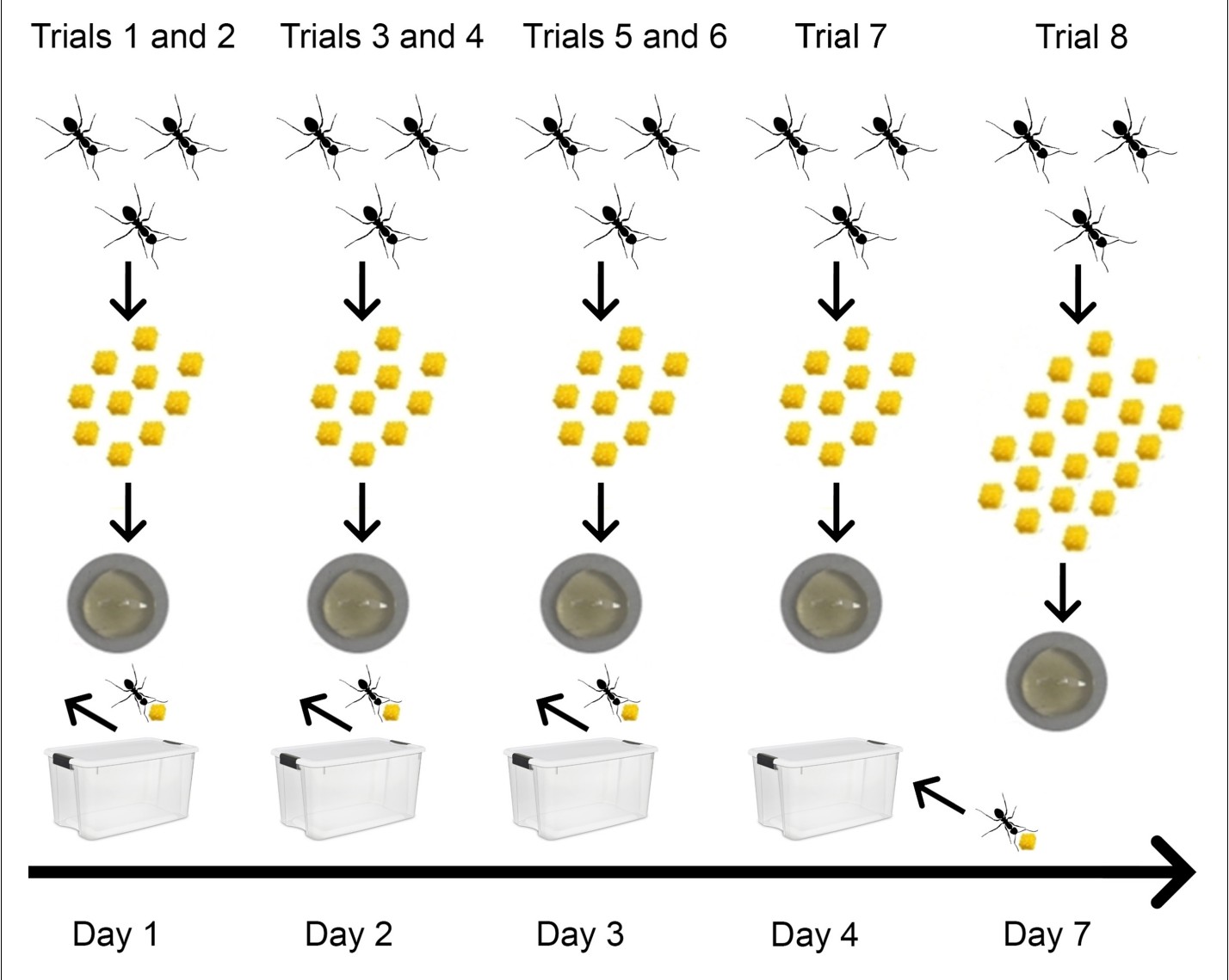

**Figure 4.** Procedure followed in experiment 3. Test-sub-colonies received honey and tools in every trial. The yellow items represent the tools (small pieces of sponge,~1 mm³), below is shown the plate with diluted honey (0.25 ml). On days 1–3, two trials were performed (10 tools offered), and after each trial, the ants that transported tools to the bait were removed. On day 4, one trial (10 tools offered) was performed and then the previously removed workers were returned to their test-sub-colony. All the workers were used in Trial 8, in which 20 tools were offered.

contrast, we found a positive correlation between the latency to find the prey and the time spent in contact with the prey: the longer the latency, the longer the time spent with the prey (session 1: $r_s$ = 0.27, $N$ = 154, p<0.001; session 2: $r_s$ = 0.42, p<0.001).

Tool users typically transported more than one tool to the bait. In most of the cases, the transport of all the 10 tools was performed by one worker, in 23% of the cases by two workers and only in one case by three workers. The removal of the active workers after each trial did not influence the characteristics of the tool transport to the bait compared to the first trial (latency of first tool to bait: LMM −0.7 < $t$ < 0.94, $N$ = 28, p>0.36; total transport time: LMM −0.7 < $t$ < 1.02, p>0.32; number of workers involved: GLMM−0.66 < $z$ < −0.07, $N$ = 28, p>0.51). Therefore, the workers that took over the tool use task were not significantly less efficient than the removed tool users.

In the final trial with 20 tools, a total of 18 workers used tools. Of these, 12 workers (66.6%) previously performed tool transport and they brought 106 (66.25%) of the 160 tools that were transported to the bait in this last trial. Of the 34 workers removed during the first 3 days of the

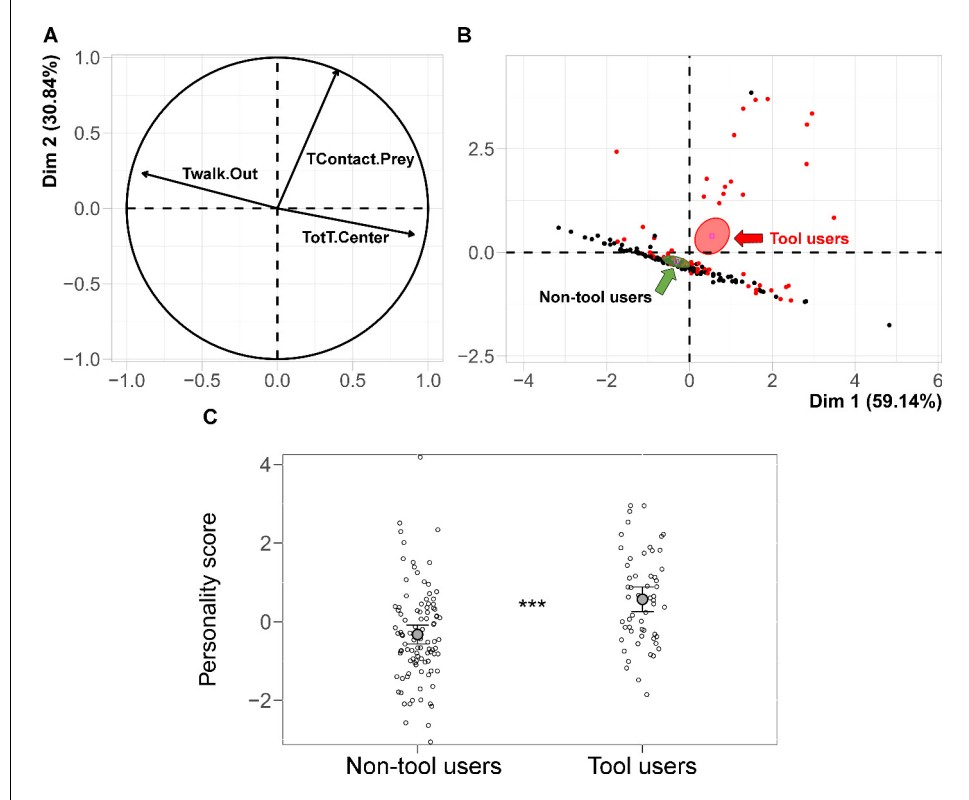

**Figure 5.** Plots of the first two dimensions of the Principal Component Analysis. Correlation circle of the variables: TWalk.Out (time spent walking in the periphery), TotT.Center (total time spent in the central area); TContact.Prey (time in contact with the prey) (A). Projection of individuals on the PCA factorial space: red dots refer to ants that used tools, black dots refer to ants that did not use tools; the confidence ellipses representing individuals using tools (red arrow) or not (green arrow) show significant difference (no overlap) (B). The probability of using tools is linked to the individual personality score. The mean and CI of the personality score and the individual data points (empty circles) for the non-tool users (mean = −0.33, CI [−0.56,−0.09]) and tool users (mean = 0.57, CI [0.26, 0.88]) is plotted (C). The ants' personality score significantly predicted the probability of using tools (***: p<0.001).

The online version of this article includes the following source data for figure 5:

**Source data 1.** Experiment 3 - the variables used in the Principal Component Analysis.
**Source data 2.** Experiment 3 - the differences in personality score between tool user and non-tool user workers.

experiment, 11 resumed tool using in this last trial, and 4 of these workers were active during the first trial with 10 tools 7 days earlier.

## Discussion

Workers of some ant species use debris as tools to collect and transport liquid food to the nest. We studied tool use behaviour at the individual level in *Aphaenogaster senilis* to determine whether any forager that knows the location of the food and the tools puts the two together, or if it is an attribute of a subset of workers. Three experiments explored this question. Experiments 1 and 3 showed that several workers contacted the food and the tools before the first tool user started transporting tools to the bait. Therefore, tool users were not simply the first workers that learned about the location of the food and the tools. Experiment 2 showed that only a small subset of foragers (about 8%) used tools and that the observed individual distribution of tool use events was not random. Moreover, tool use was a significantly repeatable behaviour across trials at the individual level. Finally, in experiment 3, we found that workers show consistent inter-individual behavioural variation in exploratory activity and reaction to prey. These highly repeatable behavioural traits were used to calculate

a personality score that significantly predicted the probability of using tools. Based solely on these behavioural traits we could distinguish two groups of workers: the tool users and the non-tool users. In sum, our data indicate that tool use is not a stochastic phenomenon but it is performed by a subset of workers with specific behavioural traits.

In both experiments 1 and 2, we observed tool users performing repeated tool transport within the same trial and also between trials. In experiment 2, more than 60% of the tool users performed repeated tool transport to the bait and their efficiency increased during the process: for a given ant, the time to transport one tool decreased with the number of tools transported and was not affected by the presence of another worker transporting tools. This improved efficiency is likely explained by the fact that ants learned the location of the tools with respect to the bait and could return quickly to pick up the next tool. In several ant species, foraging speed is enhanced by route learning (*Czaczkes et al., 2011*; *Pasquier and Grüter, 2016*), which appears to be a general phenomenon. More than 30% of the tool users participated in more than one trial, nevertheless, the performance of these workers did not improve along with the trials as the average time to transport a tool did not change significantly across trials. In other social insect species, however, an improvement over consecutive trials has been observed. For instance, bees become more efficient in foraging with experience (*Dukas and Visscher, 1994*; *Klein et al., 2019*) and, in the context of nest emigration, *Temnothorax albipennis* ant workers improve their performance over successive emigrations (*Langridge et al., 2008*). This suggests that in *A. senilis* the limiting factor to speed up the process is memorising the location of the tools, which is different in every trial.

In experiment 1, we observed that the same ant could perform both parts of the tool use task, that is transport of tools to the bait and from the bait to the nest. Therefore, as we predicted, there was no clear evidence of task partitioning. Nevertheless, our results regarding the lack of task partitioning are not conclusive due to the relatively small total number of ants that were observed participating in both parts of tool use process. To the best of our knowledge, the observations reported in the literature also lack conclusive evidence. In *A. rudis*, similarly to *A. senilis*, no evidence of task partitioning was found (*Banschbach et al., 2006*), although task partitioning was suggested in earlier observations of *A. famelica* (*Tanaka and Ono, 1978*). This suggests that this behavioural aspect might be species-specific but ideally requires further research.

The results of experiment one also showed that the dynamics of tool transport to the bait does not influence the transport of food-imbibed tools to the nest, which starts on average 2 hrs after completion of the first part. In natural conditions, covering the food quickly with debris gives an advantage to *Aphaenogaster* ants in the competition with more dominant ant species, which cannot exploit the food once it is covered (*Fellers and Fellers, 1976*), thus leaving plenty of time for *Aphaenogaster* to bring the food-imbibed tools to the nest. Therefore, it appears to be advantageous to first completely cover the food with debris and then to start transporting them to the nest. During the first part of the process, the results of experiment one showed that the higher the number of workers present in the foraging area the shorter was the latency to start the tool use process. A shorter latency resulted in a shorter total time to complete the task, but interestingly, this did not depend on the number of workers involved in tool use. Enhanced efficiency related to shorter latency was also found in the ant *Temnothorax albipennis*, where shorter recruitment latencies to high quality nest sites, compared to low-quality ones, improved the nest emigration process (*Mallon et al., 2001*; but see *Robinson et al., 2009*). Similarly, in the ant *Formica fusca*, shorter latencies to return to the nest after exploration of a novel arena characterised larger colonies and indicated efficient exploration of the surroundings (*Somogyi et al., 2020*).

Tool use events were not randomly distributed among workers, however, the results of experiment two also indicate that tool users were not necessarily specialised only in this task. About half of the workers that transported solid food (cricket legs) were also tool users. This may not hinder efficiency since it has been shown that task specialisation does not translate into higher performance in ant species without morphologically differentiated workers (*Dornhaus, 2008*). Some workers exhibited very high activity by participating in several tool use trials and transporting more than one cricket leg. These resemble 'elite' workers that show high performance in several tasks, as observed in some ant species (*Robson and Traniello, 1999*; *Pinter-Wollman et al., 2012*). It would be interesting to test whether these very active workers act as key individuals that increase the activity level of group members (*Robson and Traniello, 1999*).

In experiment 2, we also tested whether the simple fact of observing nestmates transporting tools during one trial would facilitate the task performance of naïve workers via social learning. We did not find a difference in the performance of workers that were given the possibility to observe nestmates performing tool use and workers that did not have this possibility, suggesting that social learning might not be involved in the ontogeny of the tool use behaviour in *A. senilis*. Our finding is in agreement with the observation by *Tanaka and Ono, 1978* that naïve foragers can carry out tool use in *A. famelica*. Spontaneous tool use and manufacture has been also shown in vertebrates, such as naive juvenile New Caledonian crows (*Kenward et al., 2005*), while social learning appears to be essential when the task is complex and non-natural, as observed in bumblebees (*Alem et al., 2016*; *Loukola et al., 2017*). Nevertheless, we should acknowledge that in the present study we used only one type of tool. We know that *Aphaenogaster* ants are able to select among different types of tools according to their soaking capacity and the environmental context (*Maák et al., 2017*; *Lőrinczi et al., 2018*) and that they learn to choose the optimal tools over successive trials (*Maák et al., 2017*), therefore we cannot exclude that social learning could play a role in the process of tool selection, a possibility that awaits formal testing.

Experiment 3 clearly showed that workers of *A. senilis* are characterised by consistent behavioural variation at the individual level concerning exploratory activity and reaction to prey. Consistent inter-individual behavioural variation has also been shown in other ant species (e.g. *Kühbandner et al., 2014*; *Udino et al., 2017*) and it might be a general characteristic of workers. What is new in our work is the discovery that individual behavioural variability is directly linked to the probability to perform a certain task, in this case tool use. Workers with a positive personality score, characterised by high explorative behaviour and prey-attraction, were more likely to be involved in tool use than workers with a less positive or negative personality score. Despite finding relatively small effect sizes in the analysis of the personality data, our results reveal the importance of even slight individual differences in behavioural traits, if they are consistent, in the organisation of social life. Inter-individual variability is the basis for division of labour, the phenomenon in which different individuals perform different tasks. According to the 'response threshold model' (*Robinson, 1992*; *Beshers and Fewell, 2001*), workers characterised by a low threshold will respond to low stimulus intensity (high responsiveness), while workers with high threshold will respond to high stimulus intensity (low responsiveness) (*Bonabeau et al., 1996*). Several proximate mechanisms underlying inter-individual behavioural variability have been described, including genetic diversity, ontogeny, nutrition, experience and learning (reviewed in *Jeanson and Weidenmüller, 2014*). Experience may act as 'self-reinforcement' and directly modulate the individual response threshold. It was hypothesised that the simple fact of performing a task would lower the corresponding stimulus threshold, while not performing a task would further increase the individual threshold (*Theraulaz et al., 1998*). Indeed, a study using clonal ants showed that, all else being equal, foragers that were successful had a higher probability to perform the foraging task again, compared to unsuccessful foragers, which specialised in brood care (*Ravary et al., 2007*).

Individual personality differences within a given behavioural group (foragers, nurses) may contribute to a fine-tuned division of labour but this possibility is relatively unexplored. A study in honeybees found evidence for life-long personality differences in workers and suggested that the response thresholds to some stimuli could be related to personality type, thus contributing to more robust inter-individual behavioural differences leading to division of labour even when individuals age together or share similar experiences (*Walton and Toth, 2016*). In carpenter ants, there is evidence for inter-individual variability in sucrose responsiveness and learning success in different behavioural groups of workers performing different tasks (*Perez et al., 2013*). In the ant *Myrmica rubra*, individual personality differences are connected to spatial fidelity (the position in the nest) and ants located in a given position show low thresholds to perform tasks associated with that position, thus generating division of labour (*Pamminger et al., 2014*). However, examples of interplay between personality and task performance in social insects are generally scarce.

Individual behavioural flexibility is another important aspect guaranteeing division of labour in social insects, particularly following changes in the social environment, such as changes in colony demography (*Jeanson, 2019*). We found that *A. senilis* colonies could cope well with the removal of active tool users, which were immediately replaced by individuals that were previously less active. Similarly, in the red harvester ant, *Pogonomyrmex barbatus*, when the most active foragers were experimentally removed they were replaced by other individuals (*Beverly et al., 2009*). In the ant

*Temnothorax rugatulus* as well, when the most active nurses and foragers were removed they were quickly replaced by workers from the reserve pool of inactive individuals (*Charbonneau et al., 2017*). In *T. albipennis*, in the context of nest emigration, 20% of ants were more active and performed transports repeatedly in successive experimental emigration trials (*Pinter-Wollman et al., 2012*). *Pinter-Wollman et al., 2012* performed an experiment in which they removed these active ants during some emigration trials and then they put them back. Removed active ants were replaced by previously less active individuals, but when they were returned to the colony they did not resume their active role, they were thus permanently replaced. The authors suggest that modifications in the social context and experience can cause a long-lasting change in the response threshold of workers (*Pinter-Wollman et al., 2012*; *Jeanson, 2019*). This is not what we observed in *A. senilis*, in which many of the removed active workers resumed tool use when they were returned to their sub-colony in the last trial with 20 tools, including workers that were removed 10 days earlier (after the very first trial). This is not surprising if we consider that the probability to perform tool use in *A. senilis* is linked to personality, which, by definition, is consistent over time.

In conclusion, our work shows that instead of extreme task specialisation, the involvement of workers with appropriate personality ensures high efficiency in tool use. Given the scarcity of examples linking individual personality and task performance in social insects, our study provides new insight into the interplay between personality and division of labour and should encourage further theoretical and empirical studies in this direction.

## Materials and methods

### Study species and housing

Twelve colonies of the monogynous Mediterranean ant species, *Aphaenogaster senilis*, were used. This species occurs in open and sunny locations (e.g. forest edges, lawns, fields, sand dune areas), and colony size varies between a few hundred to a few thousand workers (*Boulay et al., 2007*). Ants were collected in the Doñana National Park (Spain) in March 2019 and kept at the Laboratory of Experimental and Comparative Ethology (University Sorbonne Paris Nord, France) under standard conditions (temperature 24 ± 4°C; relative humidity 50–60%; natural light cycle). Each colony was housed in a circular plastic box (14.5 cm diameter) with a regularly moistened plaster floor (representing the nest) placed inside a larger plastic box (18 × 25.5 × 7.7 cm) constituting the foraging arena. The standard diet consisted of dead crickets (*Acheta domestica*) and apple/honey mix provided twice a week. Two weeks before the experiments, to increase the motivation for carbohydrates, the standard diet was reduced to crickets only; water was always provided ad libitum.

### Experiment 1: tool use process in whole colonies

The tool use process is composed of two parts: the transport of tools to the bait and the transport of tools from the bait to the nest (*Maák et al., 2017*). The aim of this experiment was to characterise the behaviour of ants during the entire process. We used three colonies containing the queen, brood, and around 1000 workers and we marked individually 100 workers in each colony with small dots of enamel paint. The day before the experiment, the plastic box housing the colony was connected to a separate foraging arena of the same size (18 × 25.5 × 7.7 cm) with the help of a detachable bridge allowing the workers to circulate. On the day of the experiment, 10 tools (small pieces of firm sponge, ~1 mm$^3$) were placed in this foraging arena. After 10 min, the food bait (0.25 ml of diluted honey) was placed in the arena at 5 cm distance from the tools and the bridge was removed, so that the ants present in the foraging arena with the food and the tools could not go back to the nest. This was considered as the start of the experiment. The number of ants in the foraging arena was counted and the presence of paint marked workers noted, we could thus quantify the number of workers transporting tools to the bait over the total number of workers that were present in the foraging arena. After 30 min, the connecting bridge was replaced, giving the possibility to the ants to circulate both ways and to transport tools from the bait to the nest. The experiment was repeated five times for each colony, with an interval of 1 day between two trials. Each trial was video recorded for 4 hr. We noted the latency (from the start of the experiment) of every tool dropped into the bait, the latency (from the start of the experiment) of the first tool transported from the bait to the nest

(in most of the cases, the transport of all tools inside the nest was not completed in 4 hr) and the individual identity of workers performing the task if these were marked.

## Experiment 2: is there specialisation in tool use?

In this experiment, we focused on the first part of the tool use process, the transport of tools to the bait, and we investigated whether ants specialise in tool use or whether ants that use tools to collect liquid food are also involved in the transport of solid food items (here cricket legs). We also tested the effect of a familiarisation-trial with food (diluted honey) and tools on naive workers. From each of the six source colonies (different from those used in Experiment 1), we created two sub-colonies, each with 10 larvae and 40 workers (30 foragers from outside and 10 nurses from inside the nest), individually marked with dots of enamel paint. Each sub-colony was housed in a separate plastic box (28 × 28 × 8 cm) containing a dark plastic tube with a water reservoir acting as a nest. Workers readily transported the larvae inside the nest and were left undisturbed for 1 day.

For the pre-trial, sub-colony one was familiarised with the food and the tools separately: workers received 0.25 ml of diluted honey and were allowed to feed for 15 min, then the bait was removed and, 30 min later, workers received 10 tools (pieces of sponge, ~1 mm$^3$). Sub-colony two was familiarised with the tools and the food simultaneously: workers received 0.25 ml of diluted honey and 10 tools at the same time (the tools were located 4 cm away from the honey bait). Workers were observed until all the tools were deposited on the honey bait. The two workers that interacted (i.e. antennation, grasping) most with the tools in sub-colony one and those that transported the highest number of tools in sub-colony two were removed. If observing nestmates using tools has an effect, we expected workers from sub-colonies two to be more efficient than workers of sub-colonies one in the subsequent trial.

The day after, each sub-colony was tested in three steps (*Figure 1*). *Step 1*: presentation of 0.25 ml of diluted honey and 10 tools simultaneously. Workers were observed for 45 min (during this time usually all the tools were placed on the honey bait) in which we noted the latency to transport each tool to the bait and the identity of workers using tools. *Step 2*: 2 hr later, presentation of five cricket legs (placed on the opposite side of the nest). Workers were observed for 30 min and latency to pick up each leg was noted, as well as the identity of the workers carrying the legs inside the nest. *Step 3*: 2 hr later, Step 1 was repeated (honey bait and tools).

The following day, each sub-colony was tested again with the three steps procedure. After 2 days of rest, each sub-colony received honey and 20 tools (instead of 10), to give the possibility to more workers to transport tools (*Figure 1*). We noted the latency to transport each tool to the bait and the identity of the transporting workers.

## Experiment 3: does worker personality predict tool use?

In this experiment, we investigated the possible link between personality traits and tool use. Workers were tested twice for two different personality traits and observed in tool use trials. Each sub-colony underwent seven tool use trials (*Figure 4*). After each trial, the ants that used tools were removed from the sub-colony. In this way, we could test whether personality would predict which ants will become tool users in the next trial. All tests were video-recorded. From each of the eight source colonies, we created two sub-colonies: (1) one 'test-sub-colony' with 20 individually marked workers (15 outside and five inside workers) and six larvae placed in a plastic box (25 × 18 × 9 cm) containing a dark plastic tube with a water reservoir acting as a nest; (2) one 'host-sub-colony', with 10 workers and five larvae placed in a plastic box (16 × 12 × 9 cm) with a nest tube. This host-sub-colony (2) housed the workers removed for the test-sub-colony (1) during the experiments (see below).

All the 20 workers of the test-sub-colonies (1) were individually characterised for their personality traits using two tests (open-field and reaction to prey), each repeated after a 2-day interval to assess individual consistency over time.

### Open-field

This is an adaptation of the classical open-field test developed to test exploratory behaviour and anxiety in rodents (e.g., *Prut and Belzung, 2003*) and already used with ants (*d'Ettorre et al., 2017*; *Udino et al., 2017*; *Carere et al., 2018*). An ant was individually placed in an acclimatisation tube (Ø 2.7 cm) for 1 min at the centre of a circular arena (Ø 11.5 cm) with a floor of clean filter

paper (replaced after each trial), in which an area of 9.5 cm diameter was considered as the central zone and the external part as the periphery (*Figure 3A*). Then, the tube was removed and the behaviour of the ant was observed for 3 min. More exploratory ants are expected to spend more time in the central area, while less exploratory ants will spend more time walking along the edges of the arena, where they are protected by the walls. We measured the time spent walking and resting in the central area (total time in the central area) and the time spent walking in the peripheral area with the help of a behavioural transcription software (Ethoc version 1.2, CNRS Research Centre on Animal Cognition, Toulouse).

## Reaction to prey
A similar circular arena with clean filter paper as a floor (but without the delimitation between central and peripheral zone) was used for this test. An ant was placed in the acclimatisation tube for 1 min and one fruit fly (*Drosophila hydei*) freshly killed by freezing was placed 4 cm away from the ant (*Figure 3B*). After the removal of the tube, the ant could interact with the prey for 3 min. We recorded the duration of the contact with the prey using the software Ethoc.

## Tool use
In each trial, the test-sub-colony (1) received 0.25 ml of diluted honey and 10 tools. We recorded the latency to transport each tool to the bait and the identity of workers using tools for a maximum of 45 min. Each test-sub-colony underwent seven trials: two trials on day 1 (one in the morning, one in the afternoon), two trials on day 2, two trials on day 3 and one trial on day 4 (*Figure 4*). After each trial, the ants that transported tools to the bait were removed from the test-sub-colony and placed in the host-sub-colony. In the afternoon of day 4, those workers were returned to the test-sub-colony, which had a rest of 2 days. Afterwards, a last trial was performed with 20 tools instead of 10 (*Figure 4*).

## Statistical analysis
If necessary, the variables were log-transformed prior to the analyses to meet the normality and homogeneity of residuals. Statistical analyses were carried out in R (version 3.6.1) Statistical Environment (*R Development Core Team, 2019*). In all models, the colony (or sub-colony) identity (ID) was included as a random factor.

### Experiment 1: tool use process in whole colonies
We used linear mixed-effects models (LMM, Gaussian error, maximum likelihood fit) to analyse: (i) the effect of the number of workers in the arena (before reconnecting the bridge) and of the number of workers involved in tool use on the latency of the first tool to the bait (sec); (ii) the effect of the latency of the first tool to the bait and of the number of workers involved in tool use on the total time of tool transport to the bait. In both models, the number of workers involved in tool use was included as explanatory variable, whereas the number of workers in the arena or the latency of the first tool to the bait were included as covariates. A similar model (LMM, Gaussian error, maximum likelihood fit) was built to analyse the effect of (a) latency of the first tool to the bait, (b) total transport time, (c) number of workers involved in tool use on the latency of transport the first tool to the nest. Trial number was included as a random factor. The correlation between the latency to locate the tools after contacting the food and the latency to drop the first tool into the bait was tested with Spearman rank correlation.

### Experiment 2: is there specialisation in tool use?
Number of ants using tools: to compare our empirical results to randomised data, we tested them against the null-model (any ant with sufficient information has the same probability of performing tool use) by randomly assigning the same number of observed tool use events to a simulated ant population based on the same number of individuals, sub-colonies, and trials as in experiment 2. The 95% confidence interval ($CI_{95\%}$) was calculated based on the random model with 1000 simulations performed with the help of a Macro function of Microsoft Excel (see *Supplementary file 1*).

We used the structure of the original data set of 12 sub-colonies with 40 individuals each. For 11 sub-colonies, there were four repeated measurements per individual, and in one sub-colony, there

were only three repeated measurements per individual, resulting in a total of 1880 cases (see *Supplementary file 1*). To these cases, we (a) randomly assigned 136 'tool use events', as this was the number observed in our original data set. Then, we (b) checked for each sub-colony whether each individual was assigned as a 'tool user' (one or more tool use events during the three or four repeated measurements) or not (0 tool use events). Finally (c), the number of tool users was averaged over all 12 sub-colonies, and the result was saved to a list. The whole procedure (a-c) was repeated for 1000 times and the mean number of tool users over all simulation runs (with $CI_{95\%}$) was calculated. We could then compare the observed number of tool users (mean and $CI_{95\%}$) with that generated by the model. The lack of *CI* overlap between observed and simulated data supports non-random distribution of the tool use behaviour.

Repeatability of tool use behaviour was calculated with intra-class correlations (*Lessells and Boag, 1987*) using GLMM for binomial data (tool users and non-tool users) of the R package rptR (*Stoffel et al., 2017*).

The time required for the transport of each tool during consecutive transports by the same worker within a trail was analysed with LMM (Gaussian error, maximum likelihood fit). Only workers that performed at least four consecutive transports were included. A similar model was built to compare the average time needed to transport one tool by the same worker between the trials. The order of the tools (1 to 4) or the order of the trials (up to three trials) was included as a fixed factor.

Sometimes another worker started transporting tools to the bait while the first worker was performing multiple tool transports, we thus wondered whether this 'interference' might influence the worker performance. The time needed to the first worker to transport one tool to the bait before and after interference by another worker was analysed with LMM (Gaussian error, maximum likelihood fit). The order of the tools (before and after interference) and the sub-colony ID were included as fixed factors.

The effect of the pre-trial on the (a) latency of the first tool to the bait, (b) total transport time, (c) number of workers involved in tool use in the first trial was analysed with LMMs (Gaussian error, maximum likelihood fit). The type of sub-colony was included as a fixed factor, whereas colony ID as a random factor.

## Experiment 3: does worker personality predict tool use?

Six ants died between the two sessions of the personality tests, therefore the sample size is 154 ants instead of 160 (20 ants in each of the eight sub-colonies). Repeatability across the two sessions of the open-field and the reaction to prey test was assessed with intra-class correlations (*Lessells and Boag, 1987*) by using LMM calculations (*Nakagawa and Schielzeth, 2010*) with individual identity (ID) as a random factor. The correlations between the time spent in contact with the prey (average of the two tests) and the two traits measured during the open-field test (average of the total time spent in the central area and time spent walking in the periphery) were performed with Spearman rank correlation because of the non-normal distribution of the reaction to prey data.

A 'personality score' for each ant was calculated with a Principal Component Analysis (package *Factominer*) based on (a) total time spent in the central area of the open field, (b) time spent walking in the periphery, (c) time spent in contact with the prey (average of the two tests for each variable). The personality score of a given ant is represented by its individual value of first principal component, accounting for the 59.14% of the total variance, normalised by subtracting the mean value for its colony.

The link between personality score and tool use behaviour (tool user or not) was analysed by using a GLMM with a binomial error structure (logit-link). The correlation between the latency to find the prey and the time spent in contact with the prey measured during the reaction to prey test (separately for the two sessions) was performed with Spearman rank test (data not normally distributed).

The effect of the removal of the workers on (a) latency of the first tool on bait, (b) total transport time, (c) number of workers involved in tool use was analysed with LMMs (Gaussian error, maximum likelihood fit) and GLMM (Poisson error, maximum likelihood fit), respectively, with the order of the trials as a fixed factor. In this analysis, we used data up to the first four trials because only few sub-colonies performed more than four trials.

## General details on function and packages

GLMs were performed using the *'glm'* function from the Stats package. LMMs and GLMMs were performed using *lmer* and *glmer* functions from the *lme4* package (*Bates and Maechler, 2013*). Poisson models were checked for overdispersion. The repeatability was calculated with the *rpt* function (*rptR* package, *Stoffel et al., 2017*). We assessed 95% confidence intervals ($CI_{95\%}$) by 1000 bootstraps and p values by 1000 permutations (alpha level = 0.05). For the PCA, the confidence ellipses around the categories of two factors (tool user or not) were drawn with *plotellipses* function from the *FactoMineR* package (*Le et al., 2008*).

## Acknowledgements

We thank Xim Cerdá and A de Fouchier for collecting ant colonies in Doñana National Park, Paul Devienne for technical assistance. We are grateful to Heiko G Rödel for help with the statistical simulation. Data of Experiment three are part of the Master thesis of GR. Many thanks to the Editors and three Reviewers for their very helpful comments. The study was funded by a grant of *Institut Universitaire de France* (IUF) to PdE. The first substantial revision was written during her semester residency at Washington University in St Louis as Clark Way Harrison Visiting Professor, hosted by the inspiring lab of Joan E Strassmann and David C Queller.

## Additional information

### Funding

| Funder | Grant reference number | Author |
| --- | --- | --- |
| Institut Universitaire de France | Senior Member Grant | Patrizia d' Ettorre |
| Washington University in St. Louis | Vistiting Professor Fellowship | Patrizia d' Ettorre |

The funders had no role in study design, data collection and interpretation, or the decision to submit the work for publication.

### Author contributions

István Maák, Conceptualization, Formal analysis, Supervision, Investigation, Methodology, Writing - original draft; Garyk Roelandt, Formal analysis, Investigation, Methodology, Writing - review and editing; Patrizia d'Ettorre, Conceptualization, Formal analysis, Supervision, Funding acquisition, Investigation, Methodology, Writing - original draft, Project administration, Writing - review and editing

### Author ORCIDs

István Maák (iD) http://orcid.org/0000-0002-0999-4916
Patrizia d'Ettorre (iD) https://orcid.org/0000-0001-8712-5719

### Decision letter and Author response

Decision letter https://doi.org/10.7554/eLife.61298.sa1
Author response https://doi.org/10.7554/eLife.61298.sa2

## Additional files

### Supplementary files

- Source code 1. Rmarkdown_Text.
- Source code 2. Rmarksown_Rmd.
- Source data 1. Data Experiment 1.
- Source data 2. Data Experiment 2.
- Source data 3. Data Exeperiment 3.

- Supplementary file 1. Data Simulation.
- Transparent reporting form

## Data availability

All data generated during this study are included in the manuscript and supporting files. Source data files have been provided for all Figures.

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
