## [Decision Letter]

**Acceptance summary:**

This study uses multiple experimental paradigms to investigate tool use in an ant species at both the individual and colony level. The authors show that tool use is performed by only a fraction of the non-specialized workers present. However, workers with specific personality traits, namely those that are more explorative and attracted to prey, were most likely to become tool users in the colony.

**Decision letter after peer review:**

[Editors’ note: the authors submitted for reconsideration following the decision after peer review. What follows is the decision letter after the first round of review.]

Thank you for submitting your work entitled "A small number of workers with specific personality traits perform tool use in ants" for consideration by *eLife*. Your article has been reviewed by three peer reviewers, one of whom is a member of our Board of Reviewing Editors, and the evaluation has been overseen by a Senior Editor.

Our decision has been reached after consultation between the reviewers. Although all reviewers found your manuscript interesting, they raised a large number of substantive concerns about the methodology and analyses. Additional work is required to address these points (including a comprehensive personality assay), which may fundamentally change the conclusions drawn. Based on the reviewers' extensive discussions, and their individual reviews appended below, we regret to inform you that your work, in its present form, will not be considered further for publication in *eLife*. We are willing to allow submission of a new manuscript that contains new experiments addressing the shortcomings of the present study, but please note that resubmission does not guarantee another round of in-depth review, let alone eventual acceptance.

This is a summary of the main concerns:

1) Personality claim: While this finding was the most novel, we found it to be poorly supported by the evidence presented. The two assays used to assess personality essentially only test for exploration activity, since both assays measure search/exploration time (since prey contact time is likely to be dependent on exploration, those ants who explore more will find the prey faster). In order to assess personality, we would expect more than one facet of personality to be measured, such as sociability, aggression etc. (see Udino et al., 2017). Such an approach would warrant reference to personality in this manuscript, and be much more convincing.

2) Tool-use assay: Related to the previous point, the fact that the experimental assay is also in itself an exploration assay is problematic. Without controlling for discovery or tool encounter rates, the finding can simply be explained as "those ants which are more likely to explore more are more likely to use tools". Given this confound, linking tool use to any aspect of personality is problematic.

3) Lack of clear hypotheses being presented: There is no mention of null hypotheses and predictions for the different experiments, which makes it difficult to assess results. Moreover, alternative hypotheses have not been sufficiently addressed in the manuscript (see detailed reviewer reports for more information).

4) The statistical analyses need improvement: This includes revising models where critical random or control effects appear to have been left out to ensure results are robust, adding effect sizes for all main results as well as other concerns (see reviewer reports). The authors should also include full code for all analyses rather than only the model specifications.

5) Additional concerns regarding the contextualization of the manuscript, clarity of the Materials and methods and reporting of results were brought up and can be found in the detailed reviewer reports below.

Reviewer #1:

The author's present an excellent set of experiments investigating task partitioning and specialization in one species of ants that are well known for using tools to transport food. The novel aspect of this manuscript specifically tests whether individual personality characteristics are associated with tool-use propensity, which if I understand correctly, goes beyond the usual colony approach to such questions. Overall I found the paper well written, the experiments clearly described and well designed, and the results easy to understand. The paper is likely to be of interest to a wide range of animal behavioural ecologists and I think warrants publication in *eLife*. However, I do have a couple major concerns that would need to be addressed in a revision.

1) The Introduction is generally well written but there is a lack of integration for introducing the experiments and aims of the study. The paragraph that begins “ We performed three different experiments to study the tool use behaviour…” needs to be more detailed rather than simply outlining each experiment. Can the authors include more about the overall hypotheses of the study and the predictions for the experiments? What is the aim of each experiment with respect to your overall research questions? This will also better prepare the reader for what and why particular analyses will be done for each experiment.

Essentially I think the authors can do a much better job to present the results of their experiments in a more unified and compelling way with respect to specific research questions and hypotheses.

2) The above issue, of the experiments being presented distinct and separate rather than contributing to an overall research goal needs improvement also in the Discussion. In fact parts of the Discussion would be better suited (and would be very helpful) in the Introduction. For example theoretical framing and predictions. The first two paragraphs are highly repetitive of the results but around the middle of paragraph three this improves considerably. Rather, it would be nice for the Discussion to start by briefly summarize findings across experiments with their relative contribution to the goals/hypotheses of the study.

3) My last concern includes missing random or control fixed effects in some of the mixed models used in the statistical analysis. I outline these below in detail but essentially without controlling for this additional substructure in the data the authors run the risk of producing erroneous results. Please have a look at the statistical suggestions and modify and re-run the relevant models where necessary.

Reviewer #2:

This study asks an interesting and timely question ("which individual ants perform tool use, and is tool use a personality trait?"). The experiments are performed carefully. The writing is clear. The Introduction introduces the topic well. The full data are provided in a clear manner, which is fantastic. However, I have several concerns about the conclusions drawn from the data. Firstly, some of the results on which conclusions are based are not compared against a null expectation. Secondly, and most critically, the “tool use” personality assay may not reflect anything more than tool discovery, and may thus conceptually identical to the exploration assay. The Discussion is somewhat unfocussed. In terms of subjective interest, I judge the results of experiments 1 and 2 to be not exciting enough for *eLife*. The conclusions arising from experiment 3 are exciting, but may not be supported by the data. Addressing my major concerns may require more video and statistical analysis, but will not require new experiments to be performed.

1) "Only a small number of workers performed tool use" this is technically true, but throughout the manuscript there is the implication that the tool users are in some way special (Results: "only a few workers performed the tool use behaviour"). The “null model” here would be that any ant with sufficient information has the same probability of performing tool use. Specifically, an ant must encounter both the tools and the food to perform tool use. As the results presented show, once tool use starts, almost all the tools are “used up” by one or two individuals very quickly, not giving other ants a chance to perform tool use, even if they could. The proportion of ants performing tool use is given. To put this in context, we need to know the proportion of ants which encountered tools and food at all. This “null hypothesis” needs to be explicitly presented, and refuted or not, given the data.

2) All the results presented in paragraph five are given without context of a null model, and are thus meaningless. I.e., if you randomly distributed the same number of transport events and tool use events over the same number of individuals and trials, would you get similar proportions? The empirical results can be compared to randomised data in order to clarify whether these results represent more than probabilistic behaviour amongst homogenous ants.

3) "The consecutive transport of several tools within one trial enhanced the efficiency of a tool using worker." This is interpretation, not results, and alternative null hypotheses are not excluded. It could be that when an ant “feels” like being efficient, it transports many tools and does so quickly. In other words, the “second tool” group (Figure 1B) may contain both slow ants which only manage 2 trips, and fast ants which manage 4. The “fourth tool” group only contains ants which manage at least 4 trips. Moreover, and more critically, it makes simple sense that when an ant works faster, it can manage more tools in a given time. To truly demonstrate improvement over time, only ants which manage 4 transport events should be considered, and the rank of the transport event used to predict transport speed. This may or may not have been done, the description of the analysis was not clear. The raw data is not separated by individual ant ID, so I could not check this myself.

4) The effect size of the key finding is not given. Examining the data, the normalized personality score ranges from -4 to 3 – so 7 in total. The non-tool users have an average score of -0.33. The tool users have a mean score of 0.57. I managed to recreate the analysis, and found an effect estimate of 0.6174. Using emmeans, we get a prob estimate of 0.349. This means that while the effect is highly significant, I would not say the effect size is very large.

5) I am not convinced that the “tool use” assay in experiment 3 is showing anything more than tool and food discovery probability. It stands to reason that ants which spend more time in the centre of the arena and less time on the periphery (the main components of the personality score) are more likely to encounter both the tools and the food first, and thus be in a position to being doing tool use, perhaps using up all the tools before the other workers found them, but definitely reducing the probability of other workers finding the tools (as fewer tools remain). The authors need to demonstrate that ants with a higher personality score are more likely to perform tool use even when controlling for opportunity. Otherwise, the seemingly interesting results boil down to "ants which spend more time in the centre of the arena are more likely to find things in the centre of the arena", which is not a very exciting result.

Reviewer #3:

In this manuscript, the authors perform three experiments of *Aphaenogaster senilis* using tools (e.g. debris) to soak up a liquid food resource to be carried to their nest. The primary conclusion is that a small subset of workers uses these tools, and they do so because of specific personality traits. I found the study to be quite interesting and the paper to be well-written. The authors have strong support for most of their conclusions, and indeed for the first part of their main conclusion, that only a small subset of workers perform the task, but I am less confident in the authors' evidence for the second part of that main conclusion, that these differences are due to specific personality traits.

My primary concern is in how the authors measured personality traits in workers. They performed each of two tests on two occasions: an open field test and test in which individuals could interact with prey. They found that certain behavioral traits (e.g. time spent in central arena in open field and time spent interacting with prey) were significant repeatable across the two timepoints, and conclude that these are personality traits. They then use a PCA to show that these behavioral traits correlate with tool use (ants that explored more were more likely to use tools). This consistency over time is necessary for personality, but as far as I can tell the authors did not test whether these behaviors were consistent across contexts, which is also important (I might also prefer to see more than 2 timepoints tested, though this is a minor concern). So there is a behavioral trait that is correlated across time but not contexts; I do not find this to be rigorous evidence of personality. In addition to the possibility of it being "personality" driving tool use, a non-mutually-exclusive possibility is that ants who explore more are more likely to find the tools sooner, thus becoming the tool-users, which I'm glad that the authors also discuss as a likely explanation.

I must admit that I am generally a little sceptical of the special attention sometimes placed on the idea of animal personality. To me, it is not inherently more interesting if tool use is due to a behavioral trait that is or is not consistent across contexts. Indeed, I would find the study just as interesting if the authors did not discuss personality, per se (though it is possible this would make it less broadly appealing). And I think the conclusion would be better supported and more compelling.

Other than this concern related to personality, I found the conclusions to be generally well supported. I thought it was interesting that workers improve in tool-use performance within but not across trials, and I think the authors suggestion that this relates to route-learning is a good one.

[Editors’ note: further revisions were suggested prior to acceptance, as described below.]

Thank you for submitting your article "A small number of workers with specific personality traits performs tool use in ants" for consideration by *eLife*. Your article has been reviewed by three peer reviewers, one of whom is a member of our Board of Reviewing Editors, and the evaluation has been overseen by Christian Rutz as the Senior Editor. The following individual involved in the review of your submission has agreed to reveal their identity: Tomer J Czaczkes (Reviewer #2).

The reviewers have discussed their reviews with one another, and the Reviewing Editor has drafted this decision letter to help you prepare a revised submission.

Summary:

Tool use is relatively rare in the animal kingdom and therefore of widespread academic interest. In addition to vertebrates, some social insects also demonstrate flexible tool use behaviours. In this manuscript, Maák et al. investigate task specialization, social learning and personality effects relating to tool use in the ant species *Aphaenogaster senilis*. These ants engage in tool-assisted foraging whereby debris is used as a tool to soak up liquid food, which is then transported to the nest. Previous research has shown that workers of this species can learn to use artificial materials as tools and will select the best tools based on optimal soaking characteristics.

Using a collection of three experiments, the authors demonstrate that only a small number of workers in *A. senilis* perform the tool use task but that these workers are not specialized in tool use, in that they will also carry and transport solid food items to the nest without tools. The authors found no support for social learning, specifically workers who had observed successful tool users were not more likely to become tool users themselves in subsequent trials. Remarkably, the authors did show that workers with certain personality traits were more likely to become tool users if previous tool users were removed from the colony.

Although the absolute number of workers demonstrating these phenomena was small, and further research is needed to address the repeatability and longer-term associations of worker personality and tool use behaviour, the overall contribution of this manuscript is valuable for its novel insight into tool-use behaviour in social insects at the individual level, going beyond colony dynamics.

Revisions:

The reviewers were all very content with the revised submission and congratulate the authors on substantially improving the manuscript by seriously taking into account our initial feedback. We had only a handful of major comments left that should be addressed before publication.

1) Please explain why exploring the periphery of the plate in your open field test would not in fact be ants that are the most exploratory and how this affects your inference about tool users being more explorative. Those ants who explore only in the center could actually be argued to be less explorative. Perhaps this trait needs rephrasing?

2) Can you strengthen the claim that tool users are the same individuals across trials (i.e., repeatability across trials) or at least make this more clear and transparent? A corollary of the personality-tool use conclusion is that you should have more individuals repeating tool use across multiple trials than expected by chance. This should mean that the same ants use tools over and over again. We are convinced that this is true at least within a trial, but do we see it across trials, as we should? The authors say that we do and provide evidence from experiment 1, in Table 2, and from experiment 2. However the evidence for experiments 1 is weak if we are interpreting Table 2 correctly, in total for the 3 colonies they saw only 2 ants transport tools to the bait in multiple trials (out of 19 marked tool users), and only 3 ants transport tools to the nest in multiple trials (out of 17 marker tool users). Is this actually more than would be expected by chance?

Indeed, experiment 3 provides the best evidence of personality predicting tool use across trials but given the low absolute numbers in repeat tool users across trials for Experiments 1 and 2 we would like the authors to discuss this aspect more critically in the manuscript.

3) Perhaps related to the above point, some of the results should be discussed with more transparency and caution. Specifically, can you acknowledge that effect sizes are small for the personality results and add why this small effect is still important. Please also address in the manuscript that your results for a lack of task partitioning are still not conclusive (due to the low number of ants that complete both parts of the tool use task, i.e. Table 2) and ideally requires further research.

4) Please add the latency to find prey and contact time correlation that was in the revisions letter to the reviewers but was not added to the manuscript. It would also be important to provide the correlation for all individuals, irrespective of whether they are tool users or not. Please also ensure the raw data for this correlation is included in the excel files.

---

## [Author Response]

[Editors’ note: the authors resubmitted a revised version of the paper for consideration. What follows is the authors’ response to the first round of review.]

This is a summary of the main concerns:1) Personality claim: While this finding was the most novel, we found it to be poorly supported by the evidence presented. The two assays used to assess personality essentially only test for exploration activity, since both assays measure search/exploration time (since prey contact time is likely to be dependent on exploration, those ants who explore more will find the prey faster). In order to assess personality, we would expect more than one facet of personality to be measured, such as sociability, aggression etc. (see Udino et al., 2017). Such an approach would warrant reference to personality in this manuscript, and be much more convincing.

We have clear evidence that the two personality assays do not test both for exploratory activity since it is not true that those ants who find the prey faster spend more time in contact with the prey. Under your hypothesis, we would expect that the shorter the latency to find the prey, the longer should be the contact time, i.e. a negative correlation between latency to find the prey and contact time with the prey. We had the data about latency to find the prey, so we could test these correlations. When we analyze the behavior of tool users, there is no significant correlation between latency and contact in both repeats (test 1: *r_s_* = 0.153, *p* = 0.259; test 2: *r_s_* = 0.201, *p* = 0.136, *N* = 56); when we analyze the non-tool users, there is a positive correlation (test 1: *r_s_* = 0.363, *p* = 0.0002; test 2: *r_s_* = 0.569, *p* = 0.001, *N* = 98). Therefore, we never find a correlation in the direction expected under your hypothesis. Indeed, many ants found the prey very quickly but did not show interest in it. We believe that the time spent in contact with the prey measures the interest of the ant for the prey (i.e. the motivation to bring the prey to the nest) and not the simple exploratory behavior. We did not add these data to the manuscript, but we are happy to include them if you judge it necessary.

Concerning the second point, we agree that personality has several facets and we have investigated that in a different study (Udino et al., 2017). However, for this study we believe that the two facets that we considered are sufficient and we now include a new analysis showing that there is consistency also across contexts (see reply to reviewer 3). It should be noted that the majority of the vertebrate literature considers only one trait: exploratory behavior as predictor of an individual’s personality, see for instance one of the most well-known papers in this field (more than 800 citations, google scholar) Dingemanse et al. Fitness consequences of avian personalities in a fluctuating environment. Proc R Soc B (2004). This is because exploratory tendency is often part of a behavioral syndrome including aggressiveness, neophilia, and boldness (e.g., Sih et al. Behavioural syndromes: An ecological and evolutionary overview. Trends Ecol Evol (2004)). Describing a behavioral syndrome in these ants was beyond the scope of our study.

2) Tool-use assay: Related to the previous point, the fact that the experimental assay is also in itself an exploration assay is problematic. Without controlling for discovery or tool encounter rates, the finding can simply be explained as "those ants which are more likely to explore more are more likely to use tools". Given this confound, linking tool use to any aspect of personality is problematic.

We are grateful for this comment because it forced us to analyze new data and made our study much more convincing. Tool users are not simply those ants that explore more, and thus are more likely to find the tools and the food first. We have now quantified this.

– In Experiment 1, we show that on average of about 11 workers contacted the food and the tool before the first tool user started using tools. Moreover, the first tool user was never the first worker having information about the food and the tools (see Table 3).

– In Experiment 2, we compared our data with randomized data (as suggested by reviewer 2) and we show that the observed average number of workers using tools 8.25 (*CI*_95%_ [6.47, 10.03]) is lower than the average number of tool users obtained by randomly assigning the same number of observed tool use events to a simulated ant population 10.2 (*CI*_95%_ [10.18, 10.22]). Therefore, the individual distribution of tool use events was not random in our experiments.

– In Experiment 3, we re-analyzed a subset of trials and we show that only in 5 out of 21 trials the tool user was the first worker to investigate the tools. On average, about 5 workers contacted the tools before the first tool user but did not pick them up.

More details are given in our reply to the reviewers.

3) Lack of clear hypotheses being presented: There is no mention of null hypotheses and predictions for the different experiments, which makes it difficult to assess results. Moreover, alternative hypotheses have not been sufficiently addressed in the manuscript (see detailed reviewer reports for more information).

We have clarified our hypothesis and predictions (see also reply to the reviewers).

4) The statistical analyses need improvement: This includes revising models where critical random or control effects appear to have been left out to ensure results are robust, adding effect sizes for all main results as well as other concerns (see reviewer reports). The authors should also include full code for all analyses rather than only the model specifications.

We have reanalyzed the data as suggested by the reviewers (see reply to reviewers for details) and the results did not change. Therefore, our results are robust. We had included the colony (or sub-colony) identity as a random factor, but this was not clear probably because it was written at the end of the Materials and methods section.

We now include the full codes and model outputs for all the analysis (Supplementary file 1) using the template suggested by reviewer 2.

5) Additional concerns regarding the contextualization of the manuscript, clarity of the Materials and methods and reporting of results were brought up and can be found in the detailed reviewer reports below.

We have followed all the suggestions and we modified the Introduction and Discussion. We also clarified the Materials and methods and reported more details about the results.

Reviewer #1:The author's present an excellent set of experiments investigating task partitioning and specialization in one species of ants that are well known for using tools to transport food. The novel aspect of this manuscript specifically tests whether individual personality characteristics are associated with tool-use propensity, which if I understand correctly, goes beyond the usual colony approach to such questions. Overall I found the paper well written, the experiments clearly described and well designed, and the results easy to understand. The paper is likely to be of interest to a wide range of animal behavioural ecologists and I think warrants publication in eLife. However, I do have a couple major concerns that would need to be addressed in a revision.1) The Introduction is generally well written but there is a lack of integration for introducing the experiments and aims of the study. The paragraph that begins “ We performed three different experiments to study the tool use behaviour…” needs to be more detailed rather than simply outlining each experiment. Can the authors include more about the overall hypotheses of the study and the predictions for the experiments? What is the aim of each experiment with respect to your overall research questions? This will also better prepare the reader for what and why particular analyses will be done for each experiment.Essentially I think the authors can do a much better job to present the results of their experiments in a more unified and compelling way with respect to specific research questions and hypotheses.2) The above issue, of the experiments being presented distinct and separate rather than contributing to an overall research goal needs improvement also in the Discussion. In fact parts of the Discussion would be better suited (and would be very helpful) in the Introduction. For example theoretical framing and predictions. The first two paragraphs are highly repetitive of the results but around the middle of paragraph three this improves considerably. Rather, it would be nice for the Discussion to start by briefly summarize findings across experiments with their relative contribution to the goals/hypotheses of the study.

We re-wrote parts of the Introduction, which now includes the theoretical framing and predictions, as suggested. We have clarified the aims of each experiment and linked them to the overall hypotheses. We have substantially modified the Discussion as suggested and we removed repetitive elements.

3) My last concern includes missing random or control fixed effects in some of the mixed models used in the statistical analysis. I outline these below in detail but essentially without controlling for this additional substructure in the data the authors run the risk of producing erroneous results. Please have a look at the statistical suggestions and modify and re-run the relevant models where necessary.

Thank you. We followed the suggestions.

Reviewer #2:This study asks an interesting and timely question ("which individual ants perform tool use, and is tool use a personality trait?"). The experiments are performed carefully. The writing is clear. The Introduction introduces the topic well. The full data are provided in a clear manner, which is fantastic. However, I have several concerns about the conclusions drawn from the data. Firstly, some of the results on which conclusions are based are not compared against a null expectation. Secondly, and most critically, the “tool use” personality assay may not reflect anything more than tool discovery, and may thus conceptually identical to the exploration assay. The Discussion is somewhat unfocussed. In terms of subjective interest, I judge the results of experiments 1 and 2 to be not exciting enough for eLife. The conclusions arising from experiment 3 are exciting, but may not be supported by the data. Addressing my major concerns may require more video and statistical analysis, but will not require new experiments to be performed.1) "Only a small number of workers performed tool use" this is technically true, but throughout the manuscript there is the implication that the tool users are in some way special (Results: "only a few workers performed the tool use behaviour"). The “null model” here would be that any ant with sufficient information has the same probability of performing tool use. Specifically, an ant must encounter both the tools and the food to perform tool use. As the results presented show, once tool use starts, almost all the tools are “used up” by one or two individuals very quickly, not giving other ants a chance to perform tool use, even if they could. The proportion of ants performing tool use is given. To put this in context, we need to know the proportion of ants which encountered tools and food at all. This “null hypothesis” needs to be explicitly presented, and refuted or not, given the data.

We tested the hypothesis that “any ant with sufficient information has the same probability of performing tool use” by reanalysing the videos and checking whether the tool users were the first workers to obtain information about the location of both the food and the tools in Experiment 1. We found that an average of 11.35 ± 6.49 workers (mean ± SE; min = 3, max = 33 workers), contacted the food and the tools before the first tool user dropped the first tool into bait (see Table 3 for further details). These support the conclusion that only a few workers perform the tool use behaviour. Further evidence is given below.

2) All the results presented in paragraph five are given without context of a null model, and are thus meaningless. I.e., if you randomly distributed the same number of transport events and tool use events over the same number of individuals and trials, would you get similar proportions? The empirical results can be compared to randomised data in order to clarify whether these results represent more than probabilistic behaviour amongst homogenous ants.

We compared our empirical results to randomised data. The observed average number of individuals per sub-colony showing tool use was 8.25 (*CI* [6.47, 10.03]). This was notably lower than 10.2 (*CI* [10.18, 10.22]), which is the average number of tool users obtained by randomly assigning the same number of observed tool use events to a simulated ant population based on the same number of individuals, sub-colonies, and trials, and the confidence intervals do not overlap. This indicates that the individual distribution of tool use events was not random in our experiments and therefore our results represent more than probabilistic behaviour among homogeneous ants. Moreover, the occurrences of tool use were repeatable at the individual level (*R*_ICC_ = 0.218, *p* < 0.001).

3) "The consecutive transport of several tools within one trial enhanced the efficiency of a tool using worker." This is interpretation, not results, and alternative null hypotheses are not excluded. It could be that when an ant “feels” like being efficient, it transports many tools and does so quickly. In other words, the “second tool” group (Figure 1B) may contain both slow ants which only manage 2 trips, and fast ants which manage 4. The “fourth tool” group only contains ants which manage at least 4 trips. Moreover, and more critically, it makes simple sense that when an ant works faster, it can manage more tools in a given time. To truly demonstrate improvement over time, only ants which manage 4 transport events should be considered, and the rank of the transport event used to predict transport speed. This may or may not have been done, the description of the analysis was not clear. The raw data is not separated by individual ant ID, so I could not check this myself.

We agree with the reviewer and we have redone the analysis by including only those ants that performed 4 consecutive transports. Indeed, the majority of ants in the previous dataset transported 4 tools (18 individuals out of 21, 54 vs. 60 tool transports). The results were similar to the previous ones: within trial, the time to transport one tool decreased with the number of tools transported (second-third tool: LMM *t* = -2.25, *N* = 54, *p* = 0.03, secondfourth tool: *t* = -3.75, *p* < 0.001). However, the average time to transport one tool did not change significantly between different trials (first-second: *t* = -0.015, *N* = 29, *p* = 0.99; first-third *t* = 1.02, *p* = 0.32).

We have provided the individual ant ID in the raw data.

4) The effect size of the key finding is not given. Examining the data, the normalized personality score ranges from -4 to 3 – so 7 in total. The non-tool users have an average score of -0.33. The tool users have a mean score of 0.57. I managed to recreate the analysis, and found an effect estimate of 0.6174. Using emmeans, we get a prob estimate of 0.349. This means that while the effect is highly significant, I would not say the effect size is very large.

We now give the full output of the models (Supplementary file 1). The

*CI*_95%_ of the personality score of tool users (mean = 0.57, *CI* [0.26, 0.88]) and non-tool users (mean = -0.33, *CI* [-0.56, -0.09]) does not overlap. Even if the effect size is not extremely large, we believe that our results show a clear division of personality scores between the two groups. This is also evident from the new Figure 5C, which we plotted as the reviewer suggested.

5) I am not convinced that the “tool use” assay in experiment 3 is showing anything more than tool and food discovery probability. It stands to reason that ants which spend more time in the centre of the arena and less time on the periphery (the main components of the personality score) are more likely to encounter both the tools and the food first, and thus be in a position to being doing tool use, perhaps using up all the tools before the other workers found them, but definitely reducing the probability of other workers finding the tools (as fewer tools remain). The authors need to demonstrate that ants with a higher personality score are more likely to perform tool use even when controlling for opportunity. Otherwise, the seemingly interesting results boil down to "ants which spend more time in the centre of the arena are more likely to find things in the centre of the arena", which is not a very exciting result.

We give several lines of evidence that engaging in tool use is not simply the consequence of encountering the tools. From Experiments 1 and 2 (see above) our new analyses show that tool use is not a random process performed by those workers who were the first to obtain the information about the location of the food and the tools. We also obtained new data from the videos of Experiment 3. Our results point in the direction that being more explorative is not enough for a worker to become a tool user. First, it should be noticed that the tool use process did not start immediately after the tools and the food were offered to the ants. The latency to bring the first tool to the bait was 894.71 ± 153.49 seconds (mean ± SE, N = 21 trials), which gives plenty of time for all the ants to explore and find the tools. From a total of 21 trials, only in 5 occasions the tool user was the first worker to investigate the tools. In the majority of the trials (16 out of 21) an average of about five workers investigated the tools by antennation before the first tool user but did not pick them up (4.94 ± 0.52 (mean ± SE) workers, min = 1 worker, max = 8 workers). This indicates that tool use behaviour is not merely prompted by the fact of finding the tools before other workers. As for the personality traits, we clarified that the time spent in contact with the prey is not merely linked to the exploratory activity (see our reply to the first Editors’ concern) and that the ant behaviour is consistent across contexts (see reply to reviewer 3). Moreover, tool use is repeatable at the individual level (data Experiment 2). Therefore, we believe that the relationship between tool use and personality is genuine and informative.

Reviewer #3:In this manuscript, the authors perform three experiments of Aphaenogaster senilis using tools (e.g. debris) to soak up a liquid food resource to be carried to their nest. The primary conclusion is that a small subset of workers uses these tools, and they do so because of specific personality traits. I found the study to be quite interesting and the paper to be well-written. The authors have strong support for most of their conclusions, and indeed for the first part of their main conclusion, that only a small subset of workers perform the task, but I am less confident in the authors' evidence for the second part of that main conclusion, that these differences are due to specific personality traits.My primary concern is in how the authors measured personality traits in workers. They performed each of two tests on two occasions: an open field test and test in which individuals could interact with prey. They found that certain behavioral traits (e.g. time spent in central arena in open field and time spent interacting with prey) were significant repeatable across the two timepoints, and conclude that these are personality traits. They then use a PCA to show that these behavioral traits correlate with tool use (ants that explored more were more likely to use tools). This consistency over time is necessary for personality, but as far as I can tell the authors did not test whether these behaviors were consistent across contexts, which is also important (I might also prefer to see more than 2 timepoints tested, though this is a minor concern). So there is a behavioral trait that is correlated across time but not contexts; I do not find this to be rigorous evidence of personality. In addition to the possibility of it being "personality" driving tool use, a non-mutually-exclusive possibility is that ants who explore more are more likely to find the tools sooner, thus becoming the tool-users, which I'm glad that the authors also discuss as a likely explanation.I must admit that I am generally a little sceptical of the special attention sometimes placed on the idea of animal personality. To me, it is not inherently more interesting if tool use is due to a behavioral trait that is or is not consistent across contexts. Indeed, I would find the study just as interesting if the authors did not discuss personality, per se (though it is possible this would make it less broadly appealing). And I think the conclusion would be better supported and more compelling.Other than this concern related to personality, I found the conclusions to be generally well supported. I thought it was interesting that workers improve in tool-use performance within but not across trials, and I think the authors suggestion that this relates to route-learning is a good one.

We would like to thank the reviewer for the general appreciation of our work and for the helpful comments. We formally tested the correlation between the variables collected in the exploratory behaviour assay and the reaction to prey assay and we show that they are correlated. The time spent in contact with the prey was negatively correlated with the time spent walking in the periphery (*r_s_* = -0.3, *N* = 154, *p* < 0.001) and positively correlated with the total time spent in the central area (*r_s_* = 0.33, *N* = 154, *p* < 0.001), indicating that the more exploratory ants were also those more interested in the prey. This shows that there is consistency across contexts. Therefore, we believe that we can talk about personality traits.

[Editors’ note: what follows is the authors’ response to the second round of review.]

Revisions:The reviewers were all very content with the revised submission and congratulate the authors on substantially improving the manuscript by seriously taking into account our initial feedback. We had only a handful of major comments left that should be addressed before publication.1) Please explain why exploring the periphery of the plate in your open field test would not in fact be ants that are the most exploratory and how this affects your inference about tool users being more explorative. Those ants who explore only in the center could actually be argued to be less explorative. Perhaps this trait needs rephrasing?

The open-field test is a common measure of exploratory behavior and anxiety in rodents (e.g., Prut and Belzung, 2003). The test used in our study is an adaptation of the classical open-field test. We use it to quantify exploratory activity, namely the tendency of an individual ant to thoroughly explore a novel environment. We found that this behavior is usually significantly repeatable in ants at the individual level. We used our ant open-field test in other studies, involving different ant species, to tackle different research questions. For instance, learning performance (Udino et al., 2017), cognitive judgment bias (d’Ettorre et al., 2017), individual behavioral type, and group performance (Carere et al., 2018), all cited in our manuscript. In this last study, we found that individual exploratory activity in an open-field arena was associated with the performance at the group level in cocoon recovery, thus suggesting a linear link between individual and collective behavior.

In contrast to individual ants that spend more time in the central area, ants that preferentially walk along the periphery of the open-field arena are typically looking for protection (walls) and perhaps for the possibility to escape. These ants cross the central area rapidly and therefore do not thoroughly explore the novel environment. Their possible motivation to explore is affected by their “fear” of the novel environment (please note that I use anthropomorphism here, only in order to make the picture clearer). We might even interpret this as an indication of “anxiety” in ants, analogous to rodents, but we do not (yet) dare to do so. It is also very incautious to use words such as “stress” referred to ants, but the concept is similar. Some ants are more “stressed” than others in the novel environment, therefore they explore less and spend more time at the periphery, where it is safer to walk. It is not judicious to use this phrasing in the manuscript. To be able to openly use such wording for ants (anxiety, stress, etc.), several years of research are still needed.

In the present case, *A. senilis* ants that spend more time in the central area were typically less “stressed/anxious” and showed a slow and detailed exploration of the open-field arena. These “bolder” and more explorative ants did not explore only in the center. Typically, they spent some time walking in the periphery as well, and thus exploring also this part of the arena. The so-called less explorative ants, instead, spent most of their time repeatedly walking along the edges at the periphery. Such individual differences between ants (i.e. being more explorative and “curious” towards novel objects/environments) may play a key role in relation to tool use. More explorative ants will find and approach novel tools easily, which -in turn- may result in a higher probability of using tools than less explorative ants.

We added a short explanation about the open-field in the Material and Methods:

“Open-field. This is an adaptation of the classical open-field test developed to test exploratory behaviour and anxiety in rodents (e.g., Prut and Belzung, 2003) and already used with ants (d’Ettorre et al., 2017; Udino et al., 2017; Carere et al., 2018). An ant was individually placed in an acclimatization tube (Ø 2.7 cm) for 1 min at the centre of a circular arena (Ø 11.5 cm) with a floor of clean filter paper (replaced after each trial), in which an area of 9.5 cm diameter was considered as the central zone and the external part as the periphery (Figure 3A). Then, the tube was removed and the behaviour of the ant was observed for 3 min. More exploratory ants are expected to spend more time in the central area, while less exploratory ants will spend more time walking along the edges of the arena, where they are protected by the walls.”

2) Can you strengthen the claim that tool users are the same individuals across trials (i.e., repeatability across trials) or at least make this more clear and transparent? A corollary of the personality-tool use conclusion is that you should have more individuals repeating tool use across multiple trials than expected by chance. This should mean that the same ants use tools over and over again. We are convinced that this is true at least within a trial, but do we see it across trials, as we should? The authors say that we do and provide evidence from experiment 1, in Table 2, and from experiment 2. However the evidence for experiments 1 is weak if we are interpreting Table 2 correctly, in total for the 3 colonies they saw only 2 ants transport tools to the bait in multiple trials (out of 19 marked tool users), and only 3 ants transport tools to the nest in multiple trials (out of 17 marker tool users). Is this actually more than would be expected by chance?Indeed, experiment 3 provides the best evidence of personality predicting tool use across trials but given the low absolute numbers in repeat tool users across trials for Experiments 1 and 2 we would like the authors to discuss this aspect more critically in the manuscript.

We understand the concerns of the reviewers but we do not fully agree with the above.

We believe that both Experiments 2 and 3 provide multiple and convincing evidence of repeated tool use by the same individuals across multiple trials. Experiment 2 was indeed designed to test the occurrence of repeated tool use within and across trials. We found that 33.3% of the tool using workers participated in more than one consecutive trial: “Of the 99 tool users in total, 64.7% performed repeated tool transports within the same trial and 33.3% participated repeatedly in more than one trial (Table 4). The majority of these ants (26 over 31 workers) participated in 2 trials (2.89 ± 0.42 per trial, mean ± SE), 3 workers participated in 3 trials and 2 workers in 4 trials (Table 5).” This percentage is even higher if we consider the final trial with 20 tools. In this trial, performed after one week, more than 50% of the workers showed repeated tool use across trials: “Half of the workers (50.38%) that performed tool use in the final trial with 20 tools, also participated in at least one of the previous trials with 10 tools (Table 6), thus confirming that the same ants use tools over and over again, including across trials.” In Experiment 2, we showed statistically that tool use is repeatable across trials at individual level, as stated in the previous version of the manuscript: “The occurrences of tool use were repeatable at the individual level (*R*_ICC_ = 0.218, *p* = 0.001).” The formulation of this statement was not incisive, we changed this sentence as follows: “It is important to note that the occurrences of tool use were repeatable across trials at the individual level (*R*_ICC_*=* 0.22, *N =* 1880, *p* = 0.001).” We also specified “across trials” in the corresponding sentence in the Discussion: “Moreover, tool use was a significantly repeatable behaviour across trials at the individual level.”

Repeated tool use across trials is further supported by the findings of Experiment 3, where 66.6% of workers used tools repeatedly across trials, despite the fact that they had the chance to do so only once, in the final trial with 20 tools: “In the final trial with 20 tools, a total of 18 workers used tools. Of these, 12 workers (66.6%) previously performed tool transport and they brought 106 (66.25%) of the 160 tools that were transported to the bait in this last trial. Of the 34 workers removed during the first 3 days of the experiment, 11 resumed tool using in this last trial, and 4 of these workers were active during the first trial with 10 tools seven days earlier.”

Now, to address the specific criticism (the evidence for experiments 1 is weak): Experiment 1 was an exploratory experiment in which we observed entire colonies where workers were not all individually marked. These ants have spines on the thorax and the paintmark goes away easily. Because of the presence of the spines, other methods for marking ants are difficult to use. Marking with numbers/plates on the gaster (abdomen), instead of the thorax, is not optimal since it may impede movements, particularly in this ant genus. *Aphaenogaster* (meaning: *not showing the abdomen*, from Greek: a, α privative + phaino, appearing/shining/coming to view + gaster, abdomen) is very different from *Camponotus* or *Formica*, or even *Temnothorax*, where the marking is relatively easy, also when applied on the top of the gaster. Therefore, we had to do the best of a bad job concerning individual marking in *A. senilis*. We will improve in this technical endeavor.

Contrary to Experiments 2 and 3, Experiment 1 intended to characterize the ant behavior during the two parts of the tool use process and to study the possible relationship between the number of workers using tools and the number of foragers in the arena. Observing repeated participation of the workers in the tool use process was accessory because the paint mark of the individual workers could be removed by self- and allo-grooming (as explained above). The fading of the paint marks contributed to the low number of repeated observations of tool use across trials (see Table 2). The reviewers rightly ask whether the values observed in relation to the repeated tool transport performed by the same workers across trials are higher than what would be expected by chance (by chance: 4.22). This answer is no, because these observations in Experiment 1 were affected by the fading of our pant marking. However, Table 2 was probably not easy to read, and it has been misinterpreted. In Experiment 1, there was a total of 16 marked tool users transporting tools to the bait (Table 2, column 4), and no tool users were observed to transport tools inside the nest multiple times. The number of ants transporting tools to the nest across multiple trials was confounded with the number of tool users that transported more than one tool to the nest within a single trial (column 8). We slightly modified the headings in the table to improve clarity.

3) Perhaps related to the above point, some of the results should be discussed with more transparency and caution. Specifically, can you acknowledge that effect sizes are small for the personality results and add why this small effect is still important. Please also address in the manuscript that your results for a lack of task partitioning are still not conclusive (due to the low number of ants that complete both parts of the tool use task, i.e. Table 2) and ideally requires further research.

We agree with the reviewers that Results should always be discussed with transparency and we would like to emphasize that any ambiguity that may appear in the manuscript is unintentional. We refer to the small effect sizes and their importance in the Discussion: “Despite we found relatively small effect sizes in the analysis of the personality data, our results reveal the importance of even slight individual differences in behavioural traits, if they are consistent, in the organization of social life.” We also addressed the lack of conclusive evidence regarding task partitioning in tool use and stated that this requires further research: “Therefore, as we predicted, there was no clear evidence of task partitioning. Nevertheless, our results regarding the lack of task partitioning are not conclusive due to the relatively small total number of ants that were observed participating in both parts of tool use process. To the best of our knowledge, the observations reported in the literature also lack conclusive evidence. In *A. rudis*, similarly to *A. senilis*, no evidence of task partitioning was found (Banschbach et al., 2006), although task partitioning was suggested in earlier observations of *A. famelica* (Tanaka and Ono, 1978). This suggests that this behavioural aspect might be species-specific but ideally requires further research.”

4) Please add the latency to find prey and contact time correlation that was in the revisions letter to the reviewers but was not added to the manuscript. It would also be important to provide the correlation for all individuals, irrespective of whether they are tool users or not. Please also ensure the raw data for this correlation is included in the excel files.

The required information was added to the manuscript: “If this would be the case, in the reaction to prey test we should expect a negative correlation between the latency to find prey and the time spent in contact with it (the shorter latency, the longer contact time). In contrast, we found a positive correlation between the latency to find the prey and the time spent in contact with the prey: the longer the latency, the longer the time spent with the prey (session 1: *r_s_* = 0.27, *N* = 154, *p* < 0.001; session 2: *r_s_* = 0.42, *p* < 0.001).”

The description of the analysis was also added to the Materials and methods section (Statistical analysis): “The correlation between the latency to find the prey and the time spent in contact with the prey measured during the reaction to prey test (separately for the two sessions) was performed with Spearman rank test (data not normally distributed)”. The data were included in the excel file.